# ATLAS: Constraints-Aware Multi-Agent Collaboration for Real-World Travel Planning

**Jihye Choi**[1*]**, Jinsung Yoon**[2]**, Jiefeng Chen**[2]**, Somesh Jha**[1]**, Tomas Pfister**[2]
[1]University of Wisconsin-Madison, [2]Google Cloud
{jihye,jha}@cs.wisc.edu, {jinsungyoon,jiefengc,tpfister}@google.com

## ABSTRACT

While Large Language Models (LLMs) have shown remarkable advancements in reasoning and tool use, they often fail to generate optimal, grounded solutions under complex constraints. Real-world travel planning exemplifies these challenges, evaluating agents' abilities to handle constraints that are explicit, implicit, and even evolving based on interactions with dynamic environments and user needs. In this paper, we present *ATLAS*, a general multi-agent framework designed to effectively handle such complex nature of constraints awareness in real-world travel planning tasks. ATLAS introduces a principled approach to address the fundamental challenges of constraint-aware planning through dedicated mechanisms for dynamic constraint management, iterative plan critique, and adaptive interleaved search. ATLAS demonstrates state-of-the-art performance on the TravelPlanner benchmark, improving the final pass rate from 23.3% to 44.4% over its best alternative. More importantly, our work is the first to demonstrate quantitative effectiveness on real-world travel planning tasks with live information search and multi-turn feedback. In this realistic setting, ATLAS showcases its superior overall planning performance, achieving an 84% final pass rate which significantly outperforms baselines including ReAct (59%) and a monolithic agent (27%).

## 1 INTRODUCTION

Constraint awareness and compliance is a fundamental aspect of intelligence, crucial for reasoning and problem-solving (Dechter, 2003; Holyoak & Simon, 1999). Solving real-world problems under constraints requires a delicate interplay of understanding requirements, searching for information, and synthesizing a solution that respects all rules. While Large Language Models (LLMs) have made rapid advancements in reasoning and tool use (Schick et al., 2023; Nakano et al., 2021), their reliability is still limited in practical tasks with complex, multifaceted constraints. Despite their capabilities, they often produce plans that are incoherent or invalid, a critical shortcoming for real-world deployment (Valmeekam et al., 2023; Kambhampati et al., 2024).

Existing research often sidesteps the core challenge. Some methods focus on constraint compliance but assume all necessary information is provided upfront (Parmar et al., 2025; Lee et al., 2025), while others incorporate search but presume all constraints are known in advance (Hao et al., 2025a). The more realistic and challenging scenario, where an agent must simultaneously search for context information and discover the constraints, remains largely unsolved. This intricacy is clear in a quintessential, daily task like travel planning. As shown in Figure 1, even state-of-the-art models like Gemini-2.5-Pro can satisfy a user's explicit requests (*e.g.,* budget, dates) yet fail on implicit commonsense rules, such as creating a logical itinerary or avoiding hallucinated details. The problem is further magnified in multi-turn conversations where user constraints dynamically evolve, a task where current LLMs and LLM agents still fall short (Xie et al., 2025).

To address this gap, we introduce a general multi-agent framework designed to systemically tackle three fundamental challenges in practical constraint-aware question answering tasks:

• *Constraint Construction*: Identifying the complete set of explicit and implicit constraints from user queries and search results without prior knowledge. We leverage LLMs' vast repository of

---

*This work was done while Jihye was a research intern at Google Cloud.

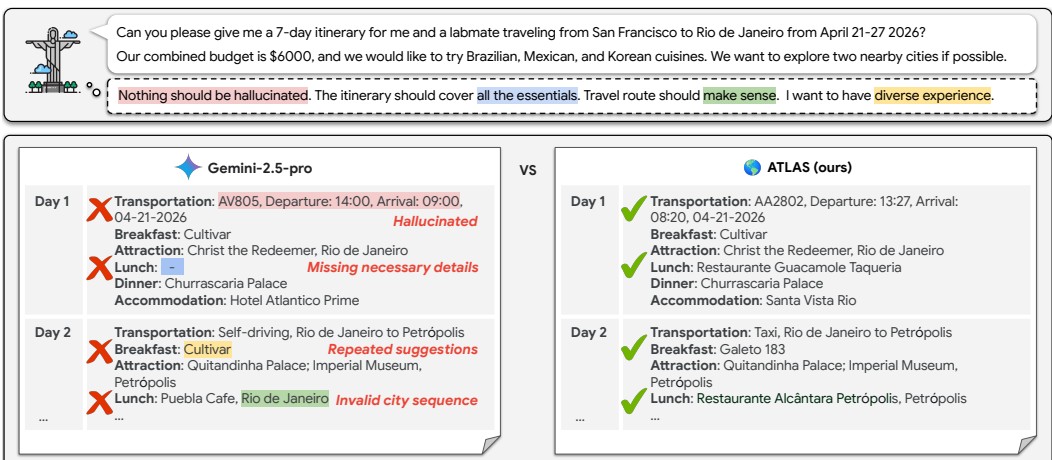

Figure 1: **Monolithic agent cannot solve real-world travel planning.** The true challenge in real-world travel planning is satisfying both explicit user requests and implicit, commonsense expectations (in dotted bubble [1]). Even advanced models like Gemini-2.5-Pro fall short, as seen in critical failures like omitting lunch after a 9 a.m. arrival or suggesting a restaurant in a different city. This highlights the vital need for a multi-agentic solution like ATLAS.

world knowledge and commonsense reasoning (Zhao et al., 2023; Krause & Stolzenburg, 2023; Ismayilzada et al., 2023) to infer and codify a full set of constraints.

- *Constraints-Aware Answering*: Generating a coherent, valid solution that is verified to adhere to all identified constraints. For this, we implement an iterative refinement loop where one agent's generation is rigorously verified against constraints by another, with targeted feedback guiding subsequent revisions (Gou et al., 2024; Choi et al., 2024).

- *Resolving Information Gap*: Diagnosing failures to determine if they stem from logical errors or insufficient information from search. We utilize an agent with high-level reasoning to diagnose the specific information gaps and recommend new search actions (Gou et al., 2025; Wu et al., 2025), effectively turning a dead-end into an opportunity to adaptively gather more information.

We demonstrate the effectiveness of our framework, ATLAS (**A**gent-based **T**ravel planning with **L**ive **A**daptive **S**earch), using travel planning tasks as a testbed, as it naturally embodies all three of these challenges. Our contributions are threefold: First, on the TravelPlanner benchmark (Xie et al., 2024), ATLAS presents superior performance than existing multi-agent baselines by up to 14% on the test set. Second, in multi-turn variants (Oh et al., 2025), ATLAS effectively adapts to evolving user feedback where other methods stagnate. Finally, we demonstrate its utility beyond sandbox settings by showing quantitative success in a live, dynamic setting that combines real-time web search with multi-turn interaction. In this realistic scenario, ATLAS achieves an 84% final pass rate with high factual grounding, while baselines like ReAct (59%) and a monolithic agent (27%) fall significantly behind.

**Related Work.** Most of existing approaches assume that all necessary information is readily available (Kambhampati et al., 2024; Yuan et al., 2025; Lee et al., 2025; Xie et al., 2025; Lu et al., 2025; Singh et al., 2024), or when integrating search tools, exhibit limited performance (Zhang et al., 2025), or lack rigorous evaluation (Chen et al., 2024). ATLAS is designed to fill these specific gaps by handling constrained travel planning with search, also with effective extension to beyond sandbox, open-domain, multi-turn setting. A detailed discussion of related work is in Appendix A.

## 2 PROBLEM SETUP

In this section, we introduce a formal definition of our target travel planning task as a generalized constrained question answering problem. In particular, given a user's natural language query $Q$, we

---

[1]The commonsense examples here are adopted from the TravelPlanner benchmark (Xie et al., 2024).

translate it into a *Constraint Satisfaction Problem* (CSP) ([Mackworth, 1977](); [Brailsford et al., 1999b]()), $P = \langle X, D, C \rangle$. The components are defined as follows:

- $X = \{x_1, x_2, \ldots, x_n\}$ is a finite set of $n$ *variables*. In travel planning, these represent the decisions to be made. For instance, $X$ includes variables such as $\texttt{Day1}_{\texttt{Transportation}}, \texttt{Day1}_{\texttt{Accommodation}}, \texttt{Day1}_{\texttt{Dinner}}, \texttt{Day2}_{\texttt{Breakfast}}$, and so on (including all relevant plan details for each day).

- $D = \{D_{x_1}, D_{x_2}, \ldots, D_{x_n}\}$ is a set of *domains*, where each $D_{x_i}$ is the finite set of possible values for variable $x_i$. These domains are dynamically constructed from external information sources rather than given a priori. That is, the agent populates $D$ from observations $O$ yielded by executing a series of tool calls. For instance, the domain for the variable $\texttt{Day1}_{\texttt{Lunch}}$ would be the set of candidate restaurants available in the relevant city on Day 1, obtained via a live search.

- $C = \{c_1, c_2, \ldots, c_m\}$ is a finite set of $m$ *constraints* that must be satisfied. Each constraint $c_j \in C$ is a pair of $\langle \text{scope}(c_j), \text{rel}(c_j) \rangle$, where $\text{scope}(c_j) \subseteq X$ is the subset of variables involved in the constraint $c_j$, and $\text{rel}(c_j) \subseteq \Pi_{x \in \text{scope}(c_j)} D_x$ is a relation specifying the allowed combinations of values for the variables in its scope. For instance, consider a constraint $c_j$ that the restaurant for all meals must be different throughout a trip. The scope would be all meal-related variables, $\text{scope}(c_j) = \{\texttt{Day1}_{\texttt{Lunch}}, \texttt{Day1}_{\texttt{Dinner}}, \texttt{Day2}_{\texttt{Breakfast}}, \ldots\}$. The relation $\text{rel}(c_j)$ would be $\text{rel}(c_j) = \{\langle r_1, \ldots, r_k \rangle \mid \forall i, j \in \{1, \ldots, k\}, i \neq j \implies r_i \neq r_j\}$, which is the set of all $k$-tuples of restaurant assignments satisfying the constraint.

Within this CSP framework, we categorize the overall constraint set $C$ into two subsets based on their source and nature (analogous to classical planning distinctions between goals and state constraints ([Fikes & Nilsson, 1971]())). (i) Explicit Constraints ($C_E$) are the requirements or preferences (*i.e.,* goals that the solution must satisfy) explicitly stated or implied by $Q$, such as budget limits, desired destination, or dates. $C_E$ could also include any new constraints that arise from observations $O$, such as accommodations requiring minimum nights stay or maximum occupancy. (ii) Implicit constraints ($C_I$) are not explicitly given but stem from commonsense domain rules and physical realities, which are analogous to the state invariants in classical planning literature. All explicit and implicit constraints together form the complete constraint set for the problem: $C = C_E \cup C_I$. The objective of the *static* travel planning problem can now be stated formally.

**Definition 2.1** (Static Travel Planning Objective). Given a CSP instance $P = \langle X, D, C \rangle$, a *complete assignment* is a function $\sigma : X \to \bigcup_{x \in X} D_x$ that maps every variable $x \in X$ to a value in its respective domain, *i.e.,* $\sigma(x) \in D_x$. A complete assignment $\sigma$ is a *feasible* solution if it satisfies all constraints in $C$. That is, for every constraint $c_j \in C$, the combination of values assigned to the variables in its scope must be an allowed tuple in its relation:

$$\langle \sigma(x) \mid x \in \text{scope}(c_j) \rangle \in \text{rel}(c_j)$$

If a feasible solution $\sigma$ exists, the problem is *satisfiable*. If no such assignment exists for the given domains $D$, the problem is *unsatisfiable*. The goal is to return a feasible solution if one is found, or an indication of unsatisfiability otherwise.

**Evolving Constraints.** The above static CSP represents a single-turn travel planning where a plan is generated in response to a one-time user query. In real-world scenarios, however, users may wish to refine the plan by providing feedback or by adding, removing, or modifying constraints in a multi-turn conversation (*e.g.,* adjusting the budget). The updated query often requires the system to gather further information, resulting in an augmented observation set and, ultimately, an evolving set of constraints. This dynamic nature transforms the problem from a static CSP into a *Dynamic CSP*, which can be viewed as a sequence of static CSPs, $P^1, P^2, \ldots, P^t$, where each problem in the sequence is a transformation of the previous one ([Mittal & Falkenhainer](); [Dechter & Dechter, 1988]()).

**Definition 2.2** (Dynamic Travel Planning Objective). Let a multi-turn conversation produce a sequence of queries $\{Q^1, Q^2, \cdots, Q^T\}$. For each turn $t \in \{1, \ldots, T\}$, the agent obtains observations $O^t$ and consequently forms a problem instance $P^t = \langle X, D^t, C^t \rangle$. A *solution trajectory* is a sequence of assignments $\{\sigma_t\}_{t=1}^T$ where each $\sigma_t$ is a feasible solution for its corresponding problem $P^t$:

$$\forall t \in \{1, \ldots, T\}, \forall c \in C^t : \langle \sigma_t(x) \mid x \in \text{scope}(c) \rangle \in \text{rel}(c)$$

Ultimately, the objective is to produce a final travel itinerary $\sigma_T$ that is feasible for $P^T$. Intermediate $\sigma^t$ serves as a (provisional) plan consistent with each intermediate problem $P^t$. This calls for a system that can manage an evolving set of constraints, generate solutions that satisfy them, and gather information according to the dynamics.

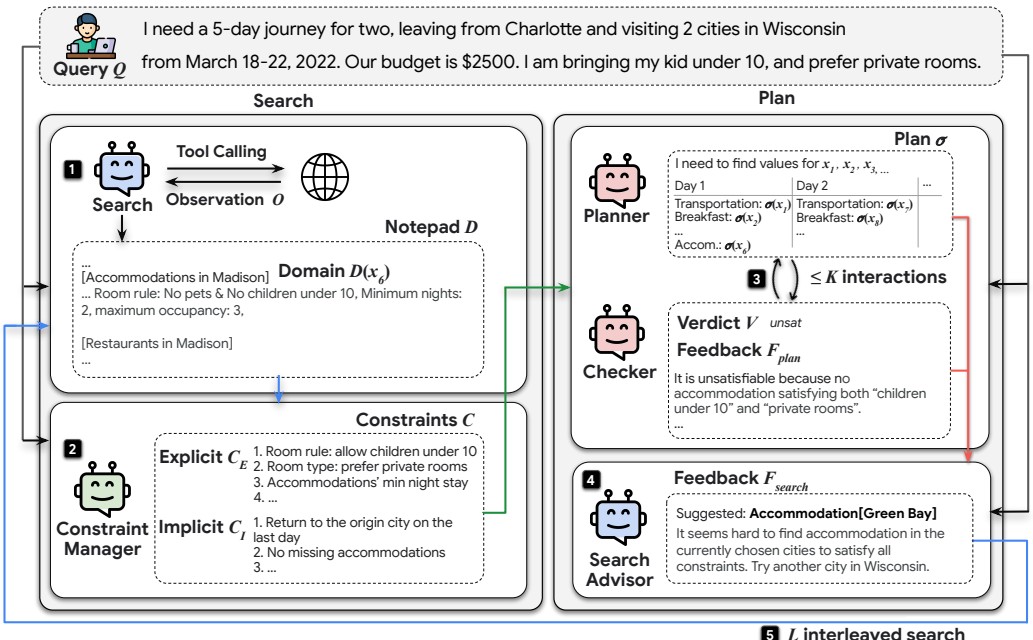

Figure 2: **An overview of our framework's workflow on a task in TravelPlanner (Xie et al., 2024).** Initially, the Search Agent populates a domain of available options, while the Constraint Manager identifies all constraints that should be considered. These include explicit constraints from the user (*e.g.,* must allow children $> 10$) and search results (*e.g.,* minimum night stays), as well as implicit, commonsense constraints. The Planner then proposes a plan, which is iteratively validated by the Checker. If the Checker finds the problem is unsatisfiable, it triggers an interleaved search. The Search Advisor diagnoses the failure and provides feedback to guide a new, more informed search.

# 3    ATLAS: AGENT-BASED TRAVEL PLANNING WITH LIVE ADAPTIVE SEARCH

**Challenges in Constraints-Aware Planning.** The design of ATLAS is directly motivated by three fundamental challenges inherent to constraint-aware planning with search. (i) Challenge 1: *Constraint construction* where the goal is to identify the complete set of implicit and explicit constraints. (ii) Challenge 2: *Planning under constraints*, reliably generating a valid plan that satisfies all identified constraint. (iii) Challenge 3: *Resolving information gaps* where the goal is to handle cases where a plan fails not due to a logical error, but due to insufficient information from the initial search. We refer the readers to Appendix B for the discussion on the connection of these challenges to the well-established planning literature in classical artificial intelligence (Brailsford et al., 1999a).

**Notations.** We now provide concrete descriptions of our framework. To model the system's operation, we use $t = 1, \ldots, T$ for the conversation turn, $\ell = 0, \ldots, L$ for the interleaved search loop within a turn, and $k = 0, \ldots, K$ for the interaction loop between the Planner and Checker agents. We use calligraphic notation to denote the space of all possible instances for these objects: $\mathcal{Q}$ for queries, $\mathcal{X}$ for variable sets, $\mathcal{O}$ for observations, $\mathcal{D}$ for domain sets, and $\mathcal{C}$ for constraint sets. Each agent is defined as a typed function (summarized in Table 3). Figure 2 and Algorithm 1 illustrate the overview.

## 3.1    CONSTRAINT MANAGEMENT (CHALLENGE 1)

The first phase of the ATLAS pipeline is to establish the factual basis for planning and is handled by two agents: The *Search Agent* is responsible for all interactions with external environments via tool calls, while the *Constraint Manager* uses the collected information to identify the constraints.

**Search (tool interaction).** Given the user query $Q^t$, it retrieves a set of raw observations $O^{t,\ell}$ from all tool calls, and then extracts a structured domain $D^{t,\ell}$ (*e.g.,* the set of relevant piece of information, such as available flights or hotels, recorded in a notepad). This process can be refined iteratively. Search Agent may receive feedback $F_{\text{search}}^{t,\ell-1} \in \mathcal{F}_{\text{search}}$ from a previous failed planning attempt, guiding

it to search for new or different information to resolve the failure (as detailed in Section 3.3).

$$\text{Search} : \mathcal{Q} \times \mathcal{F}_{\text{search}} \rightarrow \mathcal{D}, \qquad D^{t,\ell} := \text{Search}(Q^t, F_{\text{search}}^{t,\ell-1}) = (\Gamma \circ \Omega)(Q^t, F_{\text{search}}^{t,\ell-1}),$$

where $F_{\text{search}}^{t,0} = \varnothing$ by default at the start of each turn, $\Omega : \mathcal{Q} \times \mathcal{F}_{\text{search}} \rightarrow \mathcal{O}$ is the function that gathers raw observations from external environment, and $\Gamma : \mathcal{O} \rightarrow \mathcal{D}$ is the domain extraction function that filters and structures these observations into domains. We instantiate Search with standard ReAct-based tool calling module (Yao et al., 2023).

**Constraint construction.** Once the domains are populated, Constraint Manager identify and codify the complete set of constraints, $C^{t,\ell}$. This set is a combination of two types of rules: explicit constraints $C_E^{t,\ell}$ that are derived directly from the user query $Q^t$ and the current domains $D^{t,\ell}$, and implicit constraints $C_I$ that reflects fixed domain knowledge or commonsense rules that are often unstated (*e.g.,* for vacation travels, it must return to the origin city) .

$$\text{Constrain} : \mathcal{Q} \times \mathcal{D} \rightarrow \mathcal{C}, \qquad C^{t,\ell} := \text{Constrain}(Q^t, D^{t,\ell}) = C_E^{t,\ell} \cup C_I, \ C_E^{t,\ell} := \Pi(Q^t, D^{t,\ell}).$$

where $\Pi : \mathcal{Q} \times \mathcal{D} \rightarrow \mathcal{C}$ is the function for explicit-constraint extraction. We note that the Constraint Manager's role is where LLMs are particularly effective. An LLM's advanced natural language understanding allows it to expertly parse complex queries and search results to extract explicit constraints. Furthermore, its vast repository of world knowledge and commonsense reasoning enables it to infer the crucial implicit constraints that are necessary for creating a coherent and logical plan.

## 3.2 Plan Verification under Constraints (Challenge 2)

The objective of this stage is to find a valid solution for the given CSP instance, $P^{t,\ell} = \langle X, D^{t,\ell}, C^{t,\ell} \rangle$. This is addressed with an iterative loop between two specialized agents: a *Planner* and a *Checker*.

**Planning.** The Planner agent proposes a candidate solution, *i.e.,* an assignment $\sigma \in \Sigma$ where $\Sigma$ is the space of all possible assignments. It may not find a complete and valid assignment (as in Definition 2.1) at the first attempt. Hence, to improve with each attempt, its decision is informed by the history of previously failed assignments and the feedback explaining why they failed:

$$\text{Plan} : \big(\mathcal{X}, \mathcal{D}, \mathcal{C}\big) \times (\Sigma \times \mathcal{F}_{\text{plan}})^* \rightarrow \Sigma, \qquad \sigma^{t,\ell,k} := \text{Plan}\big(X, D^{t,\ell}, C^{t,\ell}; \{(\sigma^{t,\ell,i}, F_{plan}^{t,\ell,i})\}_{i=1}^{k-1}\big),$$

where $F_{plan}^{t,\ell,i} \in \mathcal{F}_{\text{plan}}$ is the feedback on the $i$th attempted planning $\sigma^{t,\ell,i}$, and initially, $F_{plan}^{t,\ell,0} = \varnothing$. Such feedback is provided by the paired Checker agent as follows.

**Constraint Checking.** The Checker agent verifies if the proposed assignment $\sigma^{t,\ell,k}$ satisfies every constraint in the set $C^{t,\ell}$. It produces two outputs: a verdict $V \in \mathcal{V}$, where $\mathcal{V} = \{\texttt{valid}, \texttt{invalid}, \texttt{unsat}\}$ and feedback $F_{\text{plan}}$ describing any unsatisfied or unsatisfiable constraints.

$$\text{Check} : \big(\mathcal{Q}, \mathcal{D}, \mathcal{C}, \Sigma\big) \rightarrow \mathcal{V} \times \mathcal{F}_{\text{plan}}, \qquad \big(V^{t,\ell,k}, F_{\text{plan}}^{t,\ell,k}\big) := \text{Check}\big(Q^t, D^{t,\ell}, C^{t,\ell}, \sigma^{t,\ell,k}\big),$$

The outcome determines the next action. If $V^{t,\ell,k} = \texttt{invalid}$, $F_{\text{plan}}^{t,\ell,k}$ is sent back to the Planner to attempt a revision $\sigma^{t,\ell,k+1}$. If $V^{t,\ell,k} = \texttt{unsat}$, the feedback indicates a deeper issue (*e.g.,* insufficient options in $D^{t,\ell}$ or incompatibilities in $C^{t,\ell}$), which triggers the next major component of our framework: an interleaved search.

## 3.3 Interleaved Search: Resolving Information Gaps (Challenge 3)

When the Checker returns an $\texttt{unsat}$ verdict, it signals that a valid plan is impossible with the current information. This triggers the *Search Advisor* agent to diagnose the underlying information gap. The Search Advisor analyzes the full context (*i.e.,* the user's query, the current domains used for planning, the constraints, and the history of failed planning attempts) to pinpoint the root cause of the failure. It then generates a targeted feedback message, $F_{\text{search}}^{t,\ell}$, guiding on what new information should be collected to make the problem satisfiable:

$$\text{SearchAdvise} : \big(\mathcal{Q}, \mathcal{D}, \mathcal{C}, (\Sigma \times \mathcal{F}_{\text{plan}})^*\big) \rightarrow \mathcal{F}_{\text{search}}, \ F_{\text{search}}^{t,\ell} := \text{SearchAdvise}(Q^t, D^{t,\ell}, C^{t,\ell}, H^{t,\ell,k}),$$

where $H^{t,\ell,k} := \{(\sigma^{t,\ell,i}, F_{plan}^{t,\ell,i})\}_{i=1}^{k}$ is the planning history so far. This task is well-suited for an LLM, which can perform high-level reasoning to diagnose the information gap and provide feedback

on search. For example, in Figure 2, as no available accommodations satisfy both the "children under 10" and "private room" constraints in the currently chosen cities, it suggests searching for options in a different city. This new search directive is then fed back to the SearchAgent in Section 3.1, which obtains an augmented domains $D^{t,\ell+1}$ and a refreshed constraint set $C^{t,\ell+1}$, continuing the loop.

## 3.4 MULTI-TURN EXTENSION

The single-turn ATLAS in Section 3 can be easily lifted to the multi-turn conversation setting with a sequence $\{Q^1, \ldots, Q^T\}$. When the user updates the query $Q^t \to Q^{t+1}$, ATLAS does not start from scratch. Instead, it uses the final domain from the previous turn, $D^{t,L}$, as a cached memory of known options. The Constraint Manager then immediately processes the new query against this cached domains to generate an updated set of constraints, $C^{t+1,1}$. That is, at the start of the $(t+1)$-th turn, the new CSP instance becomes,

$$P^{t+1,1} = \langle X, D^{t,L}, C^{t+1,1} \rangle, \qquad C^{t+1,1} := C_E^{t+1,1} \cup C_I, \quad C_E^{t+1,1} = \Pi(Q^{t+1}, D^{t,L}).$$

With this CSP, ATLAS enters the Planner-Checker loop in Section 3.2. The goal is to find a valid plan using only the information it has already gathered from the previous turn. This is a crucial efficiency step, as it avoids unnecessary tool calls if the existing knowledge is already sufficient to satisfy the user's new request. Only if this process fails—that is, if the Checker concludes with `unsat` even after $K$ revision steps—does the system determine that the cached information is insufficient. At this point, it triggers the full interleaved search process in Section 3.3, calling the Search Agent to gather new information and resuming the complete single-turn orchestration with caps $(K, L)$.

## 4 EXPERIMENTS

This section empirically validates our framework. First, we demonstrate its superior performance against competitive baselines on the standard benchmark setup along with a detailed analysis of the contribution of each core component in ATLAS (Section 4.1). Second, we extend our evaluation to multi-turn travel planning (Section 4.2). Finally, we demonstrate that ATLAS extends its superior performance to even more realistic settings with live web search and multi-turn feedback (Section 4.3). We first introduce the common experimental setup. Full implementation details are in Appendix D.1.

**Benchmark.** Our evaluations are built on the TravelPlanner benchmark (Xie et al., 2024), a standard for assessing travel planning methods in the literature (Lee et al., 2025; Kambhampati et al., 2024). It provides a sandbox environment with APIs for accommodations, restaurants, and transportation, *etc.*. This benchmark is suitable for our study because it is designed to test the capability of satisfying complex constraints under two categories: (i) *Hard constraints*, which are strict rules derived directly from the user query or search results, such as not exceeding the budget or adhering to accommodation rules; (ii) *Commonsense constraints*, are based on implicit, practical logic (see Table 4 for details).

**Evaluation Metrics.** We assess performance using the TravelPlanner benchmark's four official metrics, all reported in %: (i) Delivery rate is the percentage of queries for which *a* plan is successfully delivered. (ii) Micro pass rate is the ratio of passed constraints to total constraints considered, for both commonsense and hard constraints. (iii) Macro pass rate is the percentage of delivered plans that pass all constraints of a specific type (commonsense or hard). (iv) Final pass rate is the percentage of delivered plans that satisfy *all* commonsense and hard constraints.

## 4.1 SINGLE-TURN TRAVEL PLANNING

**Baselines.** We highlight our setup requiring agents to perform tool-based information searches and discover constraints without any prior knowledge. In the considered setting, we compare our method against three key baselines: (i) ReAct (Yao et al., 2023), the standard for tool use; (ii) Reflexion (Shinn et al., 2023) and EvoAgent (Yuan et al., 2025), popular open-sourced sole-planning baselines in the literature (Xie et al., 2024), but augmented with ReAct-based search; (iii) PMC (Zhang et al., 2025), a multi-agent framework that relies on LLM-based task decomposition and delegation. To ensure a fair comparison, all methods are limited to maximum 120 tool calls. For ATLAS, we set $K = 3$ (the maximum check steps) and $L = 10$ (the interleaved search steps). We use Gemini-2.5-Pro (Comanici et al., 2025) and Claude-Sonnet-4 (20250514) (Anthropic, 2025) as base models.

Table 1: **ATLAS consistently achieves the highest performance on the TravelPlanner benchmark.**

| Dataset | Base Model | Method | Delivery ↑ | Commonsense ↑ | | Hard Constraint ↑ | | Final Pass ↑ |
|---|---|---|---|---|---|---|---|---|
| | | | | Micro | Macro | Micro | Macro | |
| Validation (#180) | Gemini-2.5-Pro | ReAct | 98.33 | 80.32 | 29.63 | 55.55 | 46.56 | 20.56 |
| | | ReAct+Reflexion | 100.00 | 79.10 | 27.22 | 59.29 | 50.00 | 22.22 |
| | | ReAct+EvoAgent | 100.00 | 78.06 | 23.89 | 57.86 | 40.56 | 12.22 |
| | | PMC | 100.00 | 78.68 | 30.56 | 43.33 | 37.22 | 23.33 |
| | | ATLAS (ours) | **100.00** | **88.54** | **48.33** | **82.62** | **74.44** | **44.44** |
| | Claude-Sonnet-4 | ReAct | 100.00 | 79.38 | 22.78 | 56.19 | 38.89 | 11.67 |
| | | ReAct+Reflexion | 99.44 | 74.79 | 18.33 | 45.48 | 28.33 | 10.00 |
| | | ReAct+EvoAgent | 99.44 | 74.08 | 19.71 | 38.05 | 20.33 | 8.03 |
| | | PMC | 96.67 | 76.11 | 21.67 | 39.52 | 30.56 | 14.44 |
| | | ATLAS (ours) | **100.00** | **83.40** | **37.78** | **56.43** | **38.89** | **23.33** |
| Test (#1000) | Gemini-2.5-Pro | ReAct | 98.10 | 78.96 | 26.00 | 55.37 | 47.80 | 19.50 |
| | | ReAct+Reflexion | 99.90 | 77.94 | 25.90 | 65.85 | 56.70 | 22.70 |
| | | ReAct+EvoAgent | 100.00 | 78.06 | 26.23 | 60.41 | 49.23 | 15.58 |
| | | PMC | 100.00 | 79.37 | 28.30 | 57.10 | 46.10 | 21.08 |
| | | ATLAS (ours) | **100.00** | **85.81** | **40.50** | **77.64** | **70.60** | **35.00** |
| | Claude-Sonnet-4 | ReAct | 99.20 | 75.26 | 16.50 | 49.04 | 39.10 | 10.40 |
| | | ReAct+Reflexion | 99.80 | 71.84 | 13.67 | 37.84 | 26.70 | 9.13 |
| | | ReAct+EvoAgent | 98.89 | 67.01 | 10.00 | 33.71 | 20.42 | 6.11 |
| | | PMC | 100.00 | 73.89 | 15.59 | 45.19 | 33.56 | 12.12 |
| | | ATLAS (ours) | **100.00** | **78.88** | **31.00** | **49.43** | **42.00** | **18.00** |

**Results.** In Table 1, ATLAS consistently achieves the best performance across all metrics, including commonsense and hard constraint satisfaction. ATLAS outperforms PMC, another multi-agent frameworks, because its systemic orchestration—explicitly designed to handle fundamental challenges like constraint discovery, constraints-aware answering, and information gaps—is more reliable than depending on the emergent decomposition abilities of current LLMs. Furthermore, we find that simply adding search to advanced planners (*i.e.,* Reflexion, EvoAgent) yields no benefit. In fact, complex planning on poor search results can degrade performance, as seen when ReAct+EvoAgent underperforms ReAct. This failure highlights the information gap created by a single, non-adaptive search and underscores the critical importance of our interleaved search mechanism, which adaptively gathers more context as needed. We present ablation studies on travel days and task difficulty levels in Appendix D.6 and a comprehensive cost analysis of all methods in Appendix D.7.

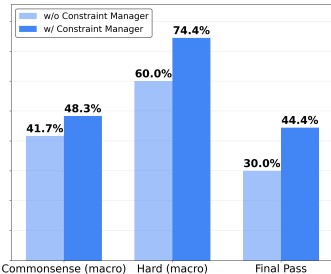 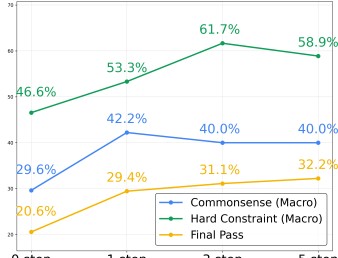 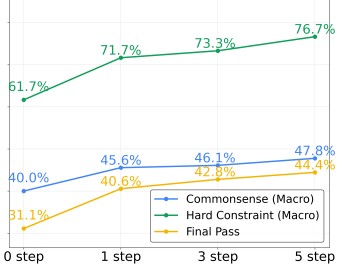

(a) Without vs with Constraint Manager during the planning phase.

(b) Varying the maximum number of revisions by Checker (K).

(c) Varying the number of interleaved search steps (L).

Figure 3: **Understanding the individual contribution of key components in ATLAS.** (a) compares the full ATLAS framework with a variant where the Constraint Manager is disabled. In (b), the baseline ($K = 0$) is a sequential search-then-plan pipeline by ReAct. In (c), the baseline ($L = 0$) is ReAct augmented with three check steps after each search. Refer to Table 5 for full results.

**Analyzing Key Components of ATLAS.** In Figure 3, we perform an ablation study to verify that each component of ATLAS successfully addresses its intended challenge. Analysis is on the validation set using Gemini-2.5-Pro. First, Figure 3a confirms the importance of explicit constraint management where disabling Constraint Manager causes 14.4% absolute drop in the macro hard constraint pass rate. Second, to improve plan validity under constraints, ATLAS decouples planning from verification using iterative revisions by Checker, and its effectiveness is evident in Figure 3b. A single check step ($K = 1$) boosts the final pass rate from 20.6% to 29.4%. However, the performance plateaus after

$K = 3$, suggesting that repeated checks yield diminishing returns when the root cause is information insufficiency from the search, making further checks inefficient [2]. This limitation highlights the need for our final component: interleaved search. In Figure 3c, activating it on a plan that has already been revised three times increases the final pass rate from 31.1% to 44.4% with $L = 5$, confirming that adaptively resolving information gaps is critical for achieving higher reliability. Collectively, these results show that each component offers a targeted and significant contribution, validating our framework's design. Additional analysis on failure cases is in Appendix D.5.

## 4.2 MULTI-TURN TRAVEL PLANNING

**Setup.** While TravelPlanner only simulates a single-turn travel planning, Flex-TravelPlanner (Oh et al., 2025) modify their validation set to simulate multi-turn travelplanning by omitting certain constraints from the original query to be introduced in subsequent turns. We follow their setup to create 2-turn and 3-turn variants of TravelPlanner benchmark, varying the type of new constraints (*local* like cuisine types or room types, or *global* like budget or the number of people) and the order in which they are introduced. We compare ReAct (*i.e.,* ours without any plan revisions or interleaved search, $K = L = 0$) and ATLAS with $K = 3, L = 10$, to validate the effectiveness of our framework in handling incremental constraints introduced over multiple turns.

Table 2: **Main results on the Flex-TravelPlanner benchmark for multi-turn planning.**

| # Turns | Constraint Type (# of samples) | Method | Delivery ↑ | Commonsense Pass ↑ | | Hard Constraint Pass ↑ | | Final Pass ↑ |
|---|---|---|---|---|---|---|---|---|
| | | | | Micro | Macro | Micro | Macro | |
| 2-Turn | + Local (189) | ReAct | 100.00 | 86.51 | 45.83 | 66.79 | 42.86 | 30.16 |
| | | ATLAS | **100.00** | **88.23** | **48.15** | **75.79** | **62.96** | **39.15** |
| | + Global (240) | ReAct | 100.00 | 85.68 | 40.00 | 68.34 | 51.67 | 26.25 |
| | | ATLAS | **100.00** | **87.55** | **48.75** | **75.97** | **64.16** | **39.59** |
| 3-Turn | + Local-then-global (378) | ReAct | 100.00 | 83.70 | 33.07 | 59.59 | 36.51 | 15.34 |
| | | ATLAS | **100.00** | **87.96** | **49.21** | **73.58** | **53.97** | **33.60** |
| | + Global-then-local (378) | ReAct | 100.00 | 84.06 | 32.80 | 59.43 | 36.24 | 17.20 |
| | | ATLAS | **100.00** | **86.81** | **47.09** | **71.38** | **52.12** | **31.75** |

**Results.** In Table 2, we observe that ATLAS consistently outperforms ReAct across all multi-turn settings. As the number of turns and complexity increases, the performance gap widens. For instance, in the 2-turn setting with local constraints introduced, we observe 9% and 13% *absolute* final pass rate gain over ReAct, when local and global constraints are introduced, respectively. As the number of turns increase to 3-turn, we observe even larger performance gain; while ReAct only shows 15.34% or 17.20 % final pass rate, ours doubles the performance regardless of the order of constraint types introduced. Full detailed results with constraint type-wise breakdowns can be found in Table 6.

## 4.3 MULTI-TURN TRAVEL PLANNING WITH LIVE SEARCH

**Setup.** To validate ATLAS's utility in a more practical setting, we evaluate it on a multi-turn travel planning with live web search. We adapt the TravelPlanner validation set by replacing its sandboxed tools with a Google Search tool, enabling agents to find real-time travel information [3]. To ensure data availability, all query dates are set to a one-month period starting from our experiment runtime (August 4th to 18th, 2025). A unique challenge of using a live search tool is the potential for the agent to hallucinate the search results. Hence, we introduce a *no hallucination rate*, a new metric measuring the percentage of plans where all information is not only derived from the search results but is also factually accurate. Factual accuracy is confirmed by an independent LLM judge equipped with its own search tool. For ATLAS , we set $K = 3, L = 5$. To simulate a realistic user interaction, we implement a multi-turn feedback loop. After an agent generates a plan, we provide corrective feedback for any constraint violations, which is added to the original query for the next turn, creating a set of evolving constraints (see Figure 4d). This is designed to model how a user collaborate with the system to *iteratively revise and improve a travel plan* which is a quite common scenario. Finally, to assess the value of our multi-agent system, we compare ATLAS against ReAct and a monolithic agent baseline that does not use multi-agent decomposition.

---

[2]In Figure 5 in the Appendix, we show that ATLAS effectively diagnoses such unsatisfiable cases and proceeds to the next interleaved search step rather than exhausting its check limits.

[3]*e.g.,* www.google.com/travel. For detailed specification in the prompts, refer to Appendix F.2

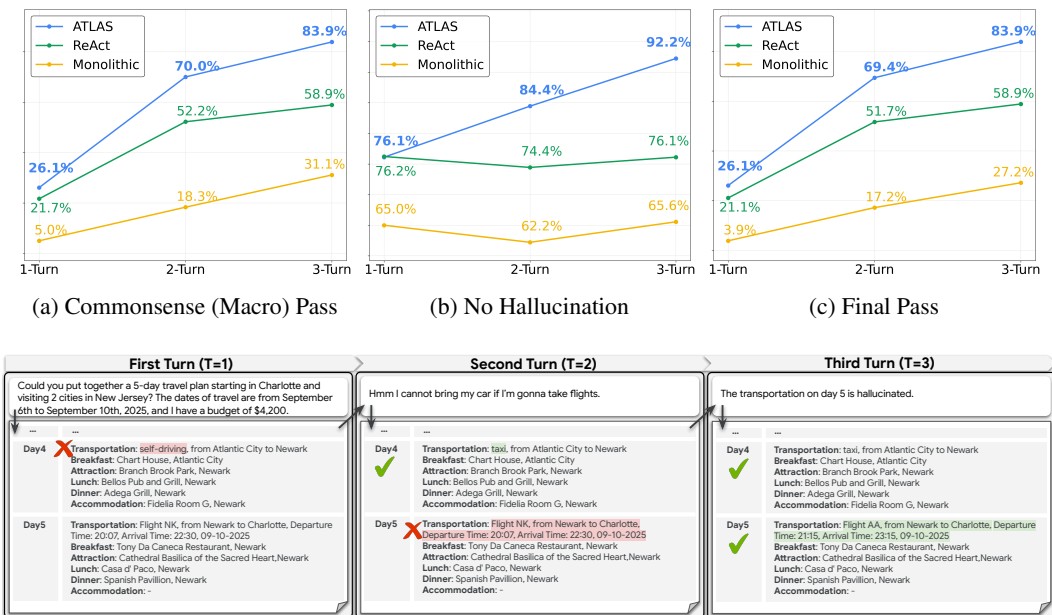

(a) Commonsense (Macro) Pass     (b) No Hallucination     (c) Final Pass

(d) Demonstration with ATLAS. End-to-end examples and the user-facing demo are in Appendix E.

Figure 4: **Live travel planning with multi-turn feedback.** See Table 7 for full results.

**Results.** Figure 4 confirm the necessity of a multi-agent approach, as the monolithic agent presents significantly lower pass rates and more hallucinations. Between ReAct and ATLAS, the performance gap is modest initially, unlike in the sandbox settings of previous sections. We attribute this to the different nature of the two environments: the original benchmark's sparse sandbox tests an agent's ability to handle information gaps and discover a single, hidden solution, whereas the live environment's abundant options make the initial task of finding any valid plan simpler, thereby reducing the immediate need for advanced search and revision capabilities. Nonetheless, the effectiveness of ATLAS emerges through the multiple turns where the user feedback acts as a series of evolving constraints. After three turns, ATLAS's final pass rate improves from 26.1% to 83.9%, while ReAct and the monolithic agent stagnate at 58.9% and 27.2%, respectively. More importantly, ATLAS effectively uses feedback to reduce factual errors, achieving a no-hallucination rate of over 92%. In contrast, ReAct's rate stagnates at 76%, showing it cannot learn from feedback to improve its factuality. This demonstrates that our proposed framework can effectively handle the evolving constraints and reliability demands of a dynamic, multi-turn planning process with live search.

## 5 CONCLUSION

In this paper, we formalized the general constrained planning tasks, and addressed its three fundamental challenges: dynamically managing a complete set of explicit and implicit rules, ensuring plan validity through iterative revision, and resolving information gaps via adaptive search. We introduced ATLAS, a multi-agent framework where specialized components systematically tackle each of these challenges. Using travel planning as a demanding testbed, ATLAS achieves state-of-the-art performance on the TravelPlanner benchmark and its multi-turn variants. Furthermore, we demonstrated its robust performance in a highly realistic setting with a live web environment and multi-turn feedback.

**Future Work.** Our work opens several exciting directions for reliable agents that can operate within the complex open-world constraints. While we used effective, simple instantiations for each component, future work could explore more advanced modules; *e.g.,* the Planner could incorporate parallel test-time scaling techniques for efficiency (Chen et al., 2025), or the Checker could be augmented with a formal CSP solver for more robust verification (Hao et al., 2025b). Beyond enhancing individual components, the framework itself could be applied to new domains requiring grounded, constraint-adherent solutions; *e.g.,* enforcing privacy policy compliance in agent guardrails (Xiang et al., 2025) or modeling personalized user preferences (Jiang et al., 2025; Li et al., 2025).

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

# APPENDIX

## A EXTENDED RELATED WORK

**Constrained question answering.** Constrained question answering is a classic planning problem where a system must process a natural language query into variables, options, and relations, then search for a feasible assignment of variables that satisfies the objective. Recent work with LLMs has advanced this capability through various approaches, including test-time schemes that amplify reasoning through sampling, self-verification, and revision (Lee et al., 2025; Chen et al., 2025), LLM-modulo paradigms that pair LLMs with external verifiers/solvers (Kambhampati et al., 2024; Gundawar et al., 2024), or multi-agent orchestration that strengthens planning via division of roles and iterative verification (Parmar et al., 2025). However, these lines largely assume *sole-planning*, where all necessary information is presumed to be available upfront (Zheng et al., 2024). They focus on improving planning capability itself, rather than the critical, real-world challenge of acquiring the necessary context information, which is often beyond the LLM's internal knowledge.

**The challenge of search in practical planning.** Real-world tasks like travel planning require both searching for viable options (*e.g.,* flights, hotels) and composing them into a constraint-consistent plan. The TravelPlanner benchmark (Xie et al., 2024) provides a realistic sandbox with tools and millions of records to test this compound competence, revealing the task's difficulty: even strong LLMs like GPT-4-Turbo achieve a near-zero final pass rate. While multi-agent frameworks show improvement, their performance still remains limited, with reported final pass rates often staying in the single digits (Xie & Zou, 2024; Yuan et al., 2025) and rarely exceeding 33% on simpler subsets of the benchmark (Zhang et al., 2025).

**Limitations of existing approaches.** There exist recent works that achieve much higher scores in this setting; for instance, a satisfiability modulo theories–backed approach reaches 93.9% with GPT-4 (Hao et al., 2025a). However, such performance hinges on having *prior, benchmark-specific knowledge* of all constraints and evaluation metrics (*e.g.,* local constraints) into the solver. Such knowledge is unavailable in practical scenarios that require *open-world discovery* is required, where an agent must dynamically extract and reconcile constraints from varied sources. Furthermore, existing work is rarely evaluated in truly realistic settings. Prior approaches are demonstrated within a sandbox, not considering *live* information search or handling *multi-turn* user interaction. While recent benchmarks like Flex-TravelPlanner (Oh et al., 2025) provide a setup with dynamic, multi-turn constraints, they still operate within a sandbox and focus on exploring sole-planning performance. The critical investigation of constraint-aware planning with live search remains largely unaddressed.

**Our positioning.** To this end, we contribute a general multi-agent framework that addresses these gaps. Our work presents reliable performance on a complex planning task without relying on prior knowledge of the constraints. Furthermore, we are the first to validate our approach in a highly practical setting that combines live information search with multi-turn conversational feedback, showing that our framework extends effectively to handle the demands of real-world travel planning.

## B    CONNECTIONS TO CLASSICAL PLANNING AND CSP LITERATURE

The design of ATLAS is motivated by challenges that are modern incarnations of well-studied problems in classical artificial intelligence, particularly in automated planning and Constraint Satisfaction Problems (CSP). Here, we elaborate on the connections between our three core challenges and this foundational literature.

**Challenge 1: Constraint Construction.** The first challenge, identifying the complete set of implicit and explicit constraints, mirrors the classic knowledge-acquisition bottleneck. In the CSP literature, this is known as constraint acquisition, a line of work focused on learning or eliciting missing constraints that are not explicitly stated in the initial problem definition. Early systems explored interactive methods to acquire constraints from users, while later work developed techniques to automatically learn them from examples or interaction, which is analogous to how ATLAS must infer rules from a natural language query and search results (Freuder & Wallace, 1998; Bessiere et al., 2007; Rossi & Sperduti, 2004).

**Challenge 2: Constraints-Aware Planning.** The second challenge, generating a valid plan that verifiably adheres to all known rules, follows a long-established principle in automated planning. Classical systems often relied on a strict separation between a plan synthesizer (the generator) and a plan validator (the checker). A plan would be generated and then formally validated against a world model and a set of rules, often expressed in languages like PDDL (Fox & Long, 2003; Howey et al., 2004). Our Planner-Verifier architecture directly implements this robust "generate-then-test" paradigm, a technique now being adapted to improve the reliability and conformance of modern LLMs (Gou et al., 2024; Choi et al., 2024).

**Challenge 3: Resolving Information Gap.** The third challenge addresses failures caused by insufficient information rather than logical errors. This problem was central to classical planning in partially observable environments, where agents had to interleave planning with sensing actions to gather new information about the world before proceeding (Petrick & Bacchus, 2002; 2004). This concept also drove the development of mixed-initiative systems, where the most effective approach involves alternating between proposing partial plans and fetching missing facts, often in collaboration with a human user (Ferguson et al., 1996; Hearst, 1999). ATLAS's adaptive interleaved search is a direct implementation of this principle, enabling the agent to diagnose and resolve information gaps on the fly.

# C  ADDITIONAL DESCRIPTIONS OF ATLAS

| Function (Agent) | Input types | Output types | Role |
|---|---|---|---|
| Search | $Q^t$, $F_{\text{search}}^{t,\ell-1}$ | $D^{t,\ell}$ | Tool interaction & domain construction |
| Constrain | $Q^t$, $D^{t,\ell}$ | $C^{t,\ell}$ | Identify explicit & implicit constraints |
| Plan | $P^{t,\ell}$, $\{(\sigma^{t,\ell,i}, F_{\text{plan}}^{t,\ell,i})\}_{i=1}^{k-1}$ | $\sigma^{t,\ell,k}$ | Propose a candidate plan |
| Check | $Q^t$, $P^{t,\ell}$, $\sigma^{t,\ell,k}$ | $V^{t,\ell,k}$, $F_{\text{plan}}^{t,\ell,k}$ | Verify plan; provide feedback on violations |
| SearchAdvise | $Q^t$, $P^{t,\ell}$, $\{(\sigma^{t,\ell,i}, F_{\text{plan}}^{t,\ell,i})\}_{i=1}^{k}$ | $F_{\text{search}}^{t,\ell}$ | Diagnose unsatisfiability & suggest new search |

Table 3: Typed function signatures for the agents in ATLAS. We use $P^{t,\ell}$ as a shorthand for the CSP instance $\langle X, D^{t,\ell}, C^{t,\ell} \rangle$. The indices $t, \ell, k$ represent the conversation turn, the (interleaved) search step, and the planner-checker interaction step, respectively.

We provide the algorithmic overview of ATLAS in Algorithm 1.

---

**Algorithm 1** Multi-Turn Travel Planning with ATLAS

---

**Require:** Sequence of user queries $\{Q^1, \dots, Q^T\}$; Max loops $(L, K)$
1: $D_{\text{cache}} \leftarrow \emptyset$         ▷ Initialize domain memory
2: **for** $t = 1$ to $T$ **do**         ▷ Loop over conversation turns
3:     $D^{t,1} \leftarrow D_{\text{cache}}$         ▷ Start with cached domain from the previous turn
4:     **for** $\ell = 1$ to $L$ **do**         ▷ Loop for interleaved search
5:        $C^{t,\ell} \leftarrow \text{Constrain}(Q^t, D^{t,\ell})$
6:        $P^{t,\ell} \leftarrow \langle X, D^{t,\ell}, C^{t,\ell} \rangle$
7:        $history_{\text{plan}} \leftarrow [\,]$
8:        $V \leftarrow$ invalid         ▷ Initialize verdict for this search loop
9:        **for** $k = 1$ to $K$ **do**         ▷ Inner loop for plan-and-check attempts
10:           $\sigma \leftarrow \text{Plan}(P^{t,\ell}, history_{\text{plan}})$
11:           $(V, F_{\text{plan}}) \leftarrow \text{Check}(Q^t, P^{t,\ell}, \sigma)$
12:           $history_{\text{plan}}.append((\sigma, F_{\text{plan}}))$
13:           **if** $V =$ valid **then**
14:              $D_{\text{cache}} \leftarrow D^{t,\ell}$         ▷ Update memory with the successful domain
15:              **break**         ▷ Exit inner and outer loops for this turn
16:           **else if** $V =$ unsat **then**
17:              **break**         ▷ Exit inner loop to trigger a new search
18:           **end if**
19:        **end for**
20:        **if** $V =$ valid **then**
21:           Output solution $\sigma$ and proceed to next turn.
22:           **break**
23:        **else**         ▷ If no solution found, diagnose failure to guide the next search
24:           $F_{\text{search}} \leftarrow \text{SearchAdvise}(Q^t, P^{t,\ell}, history_{\text{plan}})$
25:           $D^{t,\ell+1} \leftarrow \text{Search}(Q^t, F_{\text{search}})$         ▷ Get new domain for the next iteration
26:        **end if**
27:     **end for**
28: **end for**

---

# D    ADDITIONAL EXPERIMENTS

## D.1    ADDITIONAL DETAILS ON THE EXPERIMENTAL SETUP

**TravelPlanner (Xie et al., 2024).** This benchmark includes 180 queries for validation set, and 1000 queries for the test, where each query specifies explicit requests (*e.g.,* trip duration, budget, cuisine preferences, etc.) that define the constraints the final plan must satisfy. These are categorized into: (i) *Hard constraints*, which are strict rules derived directly from the user query, such as not exceeding the budget or adhering to a required cuisine types; (ii) *Commonsense constraints*, are based on implicit, practical logic, such as ensuring a reasonable route during the given trip duration, or planning a day's activities in the same city. This benchmark provides a sandbox environment including accommodations, restaurants, and transportation, and the agent is expected to search information relevant to the query within the sandbox. Detailed descriptions on the constraint types are in Table 4.

Table 4: **Descriptions on the considered constraint.**

| Legend in Figures | Description |
|---|---|
| *Commonsense Constraint* | |
| Conflicting Transportations | Transportation choices within the trip must be reasonable. For example, having both "self-driving" and "flight" would be considered a conflict. |
| Incomplete Information | No key information should be left out of the plan (*e.g.,* lack of accommodation) |
| Unreasonable City Route | Changes in cities during the trip must be reasonable. |
| < Minimum Nights Stay | The number of consecutive days spent in a specific accommodation during the trip must meet the corresponding required minimum number of nights' stay |
| Repeated Restaurants | Restaurant choices should not be repeated throughout the trip. |
| Hallucinated Details | All information in the plan must be within the closed sandbox; otherwise, it will be considered a hallucination. |
| Outside the Current City | All scheduled activities for the day must be located within that day's city(s). |
| *Hard Constraint* | |
| Eval unqualified | Checking hard constraint is not meaningful since some details are missing or hallucinated. |
| Budget exceeded | The total budget of the trip |
| Cuisine | Cuisines include "Chinese", "American", "Italian", "Mexican", "Indian", "Mediterranean", and "French". |
| Room Rule | Room rules include "No parties", "No smoking", "No children under 10", "No pets", and "No visitors" |
| Room Type | Room types include "Entire Room", "Private Room", "Shared Room", and "No Shared Room" |

**TravelPlanner with Live Search.** We adapt the TravelPlanner benchmark to assess performance in a realistic, open-domain setting. While retaining the queries from the validation set, we replace the benchmark's static, sandboxed data for accommodations, flights, restaurants, and attractions with a live Google Search tool. This required modifying certain unrealistic constraints present in the original data. For instance, accommodation attributes such as "Room type" (*e.g.,* "private room") and "Room rule" (*e.g.,* "No parties") are not typically available through real-world search. Consequently, we excluded these specific constraints from both our information-gathering prompts and the final evaluation criteria to ensure a fair assessment under real-world conditions.

**Additional Evaluation Metric in TravelPlanner with Live Search.** We note that live search introduces a challenge not present in the sandbox setting: the search results themselves can be hallucinated by the search agent. Hence, in addition to the evaluation metrics used in the sandbox setting (Section 2), we introduce an additional metric to capture this aspect: *no hallucination rate*. It measures the ratio of plans with all travel details being drawn from the *not hallucinated* search results, to total plans attempted. To enable this, we require the search agent to output retrieved information in a structured format, exactly the same as the sandbox data format for flights, accommodations, restaurants, and attractions, and we use a separate LLM-based judge (with the same base model) to verify whether each travel detail in the final plan exists on the web.

## D.2 DETAILED RESULTS ON ABLATION STUDY

Table 5: **Ablations on the key components of ATLAS .** Results are on the TravelPlanner validation set using Gemini-2.5-pro.

(a) Without vs with Constraint Manager using 5 check steps and 10 interleaved search steps.

|  | Delivery | Commonsense | | Hard Constraint | | Final Pass |
|---|---|---|---|---|---|---|
|  |  | Micro | Macro | Micro | Macro |  |
| w/o Constraint Manager | 100.00 | 86.41 | 41.67 | 73.21 | 60.00 | 30.00 |
| w/ Constraint Manager | 100.00 | 88.54 | 48.33 | 82.62 | 74.44 | 44.44 |

(b) Varying the maximum number of verificataion steps by Checker. Baseline is the sequential search-then-planning by ReAct.

| # Steps | Delivery | Commonsense | | Hard Constraint | | Final Pass |
|---|---|---|---|---|---|---|
|  |  | Micro | Macro | Micro | Macro |  |
| Baseline | 95.56 | 77.36 | 26.11 | 47.14 | 38.89 | 17.78 |
| + 1 step | 100.00 | 85.42 | 42.22 | 64.76 | 53.33 | 29.44 |
| + 3 steps | 100.00 | 84.79 | 40.00 | 73.81 | 61.67 | 31.11 |
| + 5 steps | 100.00 | 84.65 | 40.00 | 68.33 | 58.89 | 32.22 |

(c) Varying the number of interleaved search steps. Baseline is ReAct augmented with three critiques steps after each search.

| # Steps | Delivery | Commonsense | | Hard Constraint | | Final Pass |
|---|---|---|---|---|---|---|
|  |  | Micro | Macro | Micro | Macro |  |
| Baseline | 100.00 | 84.79 | 40.00 | 73.81 | 61.67 | 31.11 |
| + 1 step | 100.00 | 86.81 | 45.56 | 80.71 | 71.67 | 40.56 |
| + 3 steps | 100.00 | 87.57 | 46.11 | 87.57 | 73.33 | 42.78 |
| + 5 steps | 100.00 | 88.12 | 47.78 | 84.76 | 76.67 | 44.44 |

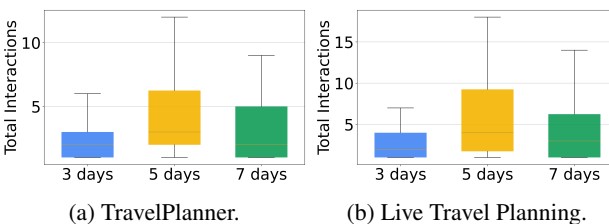

(a) TravelPlanner.  (b) Live Travel Planning.

Figure 5: **Distribution of the total number of check steps.** We set maximum three critique steps per each search step including 10 additional interleaved search steps.

As shown in Figure 5, even when we allow for a high maximum number of interactions (*i.e.,* 33 steps), ATLAS resolves most cases within 15 total interactions between Planner and Checker. This indicates that ATLAS is mostly efficient at finding a valid solution before exhausting its limits. Based on these trends, a strategic tuning of these hyperparameters is crucial for balancing performance and cost in a real-world deployment.

### D.3  FULL RESULTS ON MULTI-TURN TRAVEL PLANNING

**Flex-TravelPlanner.** In addition to Table 2, we provide the detailed results on Flex-TravelPlanner benchmark in Table 6.

Table 6: **Results of ATLAS on the Flex-TravelPlanner benchmark.**

| Constraint Type | Method | Delivery ↑ | Commonsense ↑ | | Hard Constraint ↑ | | Final Pass ↑ |
|---|---|---|---|---|---|---|---|
| | | | Micro | Macro | Micro | Macro | |
| *2-Turn (Local)* | | | | | | | |
| + Cuisine (#48) | ReAct | 100.00 | 79.43 | 31.25 | 70.95 | 47.92 | 22.92 |
| | ATLAS | 100.00 | 85.42 | 41.67 | 85.81 | 75.00 | **35.42** |
| + Room rule (#77) | ReAct | 100.00 | 87.66 | 48.85 | 64.55 | 36.36 | 29.87 |
| | ATLAS | 100.00 | 89.29 | 51.95 | 66.79 | 53.25 | **38.96** |
| + Room type (#64) | ReAct | 100.00 | 90.43 | 53.12 | 66.36 | 46.88 | 35.94 |
| | ATLAS | 100.00 | 89.06 | 48.44 | 79.09 | 65.62 | **42.19** |
| *2-Turn (Global)* | | | | | | | |
| + Number of People (#120) | ReAct | 100.00 | 86.35 | 42.50 | 70.00 | 51.67 | 25.83 |
| | ATLAS | 100.00 | 87.50 | 50.83 | 77.50 | 65.83 | **41.67** |
| + Budget (#120) | ReAct | 100.00 | 85.00 | 37.50 | 66.67 | 51.67 | 26.67 |
| | ATLAS | 100.00 | 87.60 | 46.67 | 74.44 | 62.50 | **37.50** |
| *3-Turn* | | | | | | | |
| + Local-then-global (#378) | ReAct | 100.00 | 83.70 | 33.07 | 59.59 | 36.51 | 15.34 |
| | ATLAS | 100.00 | 87.96 | 49.21 | 73.58 | 53.97 | **33.60** |
| + Global-then-local (#378) | ReAct | 100.00 | 84.06 | 32.80 | 59.43 | 36.24 | 17.20 |
| | ATLAS | 100.00 | 86.81 | 47.09 | 71.38 | 52.12 | **31.75** |

**Multi-Turn Travel Planning with Live Search.** In addition to Figure 4, we provide the full results on all metrics in Table 7.

Table 7: **ATLAS Multi-turn feedback on live travelplanning.**

| Method | Delivery ↑ | Commonsense ↑ | | Hard Constraint ↑ | | No Hallucination ↑ | Final Pass ↑ |
|---|---|---|---|---|---|---|---|
| | | Micro | Macro | Micro | Macro | | |
| Monolithic | 81.11 | 61.60 | 5.00 | 10.04 | 10.00 | 65.00 | 3.89 |
| + 1-turn | 85.56 | 72.29 | 18.33 | 22.58 | 22.78 | 62.22 | 17.22 |
| + 2-turn | 90.00 | 79.79 | 31.11 | 29.75 | 27.78 | 65.56 | 27.22 |
| ReAct | 100.00 | 80.28 | 21.67 | 50.54 | 51.67 | 76.22 | 21.11 |
| + 1-turn | 100.00 | 91.18 | 52.22 | 62.01 | 62.78 | 74.44 | 51.67 |
| + 2-turn | 99.44 | 92.15 | 58.89 | 67.38 | 67.22 | 76.11 | 58.89 |
| ATLAS | 100.00 | 82.08 | 26.11 | 64.87 | 62.78 | 76.11 | **26.11** |
| + 1-turn | 100.00 | 94.79 | 70.0 | 77.06 | 75.56 | 84.44 | **69.44** |
| + 2-turn | 100.00 | 96.81 | 83.89 | 86.38 | 86.11 | 92.22 | **83.89** |

## D.4 Without vs With Unconventional Benchmark-specific Constraints

In addition to Figure 6, we show the full results for all evaluation metrics in Table 8.

Table 8: **ATLAS without vs with benchmark-specific unconventional requirements directly provided to the Constraint Manager.**

| Base Model | Constraint Manager | Delivery ↑ | Commonsense ↑ | | Hard Constraint ↑ | | Final Pass ↑ |
|---|---|---|---|---|---|---|---|
| | | | Micro | Macro | Micro | Macro | |
| Gemini-2.5-pro | without hint | 100.00 | 88.54 | 48.33 | 82.62 | 74.44 | 44.44 |
| | with hint | 100.00 | 93.61 | 70.56 | 79.29 | 71.11 | 59.44 |
| Claude-Sonnet-4 | without hint | 100.00 | 83.40 | 37.78 | 56.43 | 38.89 | 23.33 |
| | with hint | 100.00 | 85.42 | 48.33 | 60.71 | 47.22 | 33.33 |

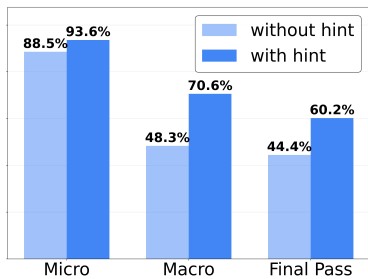

Figure 6: Providing the two benchmark-specific rules as constraints to the Constraint Manager further improves the performance, demonstrating our framework's ability to effectively use explicit guidance.

### D.5 FAILURE ANALYSIS OF ATLAS

**TravelPlanner.** Constraint failure analysis reveals ATLAS's effectiveness and the nature of its remaining errors (Figure 7). Compared to the simplest baseline with $K = L = 0$, ATLAS substantially reduces failures across all categories, especially in eliminating hallucinations and including necessary details (see light green and green bars in Figure 7a vs. 7b). It excels at enforcing constraints discovered from search results (*e.g.,* an accommodation's "minimum nights stay" rule from 33 to 4). This success stems from the Constraint Manager, which reasons over and identifies emergent rules from both the user query and retrieved data.

Despite these gains, two error types remained prominent: "Conflicting Transportations" and "Repeated Restaurants". These reflect benchmark-specific rules, termed *unconventional hints* in Zhang et al. (2025), that may not necessarily align with real-world practices (*e.g.,* a user cannot visit the same restaurant twice, or mix flights with self-driving). Adopting the setup by Zhang et al. (2025) that provide them as additional hints to their planning agent, we consider providing these as explicit constraints to our Constraint Manager. This change cause the final pass rate to jump from 44% to 60% in Figure 6, and the corresponding failure analysis (Figure 7c) confirms that these two error types were largely eliminated, and now remaining errors primarily due to the base model's inherent reasoning limitations, such as creating illogical city routes or still hallucinating some details. This demonstrates that ATLAS is highly effective at ensuring constraint compliance, pushing performance to the limits of the base model's capabilities.

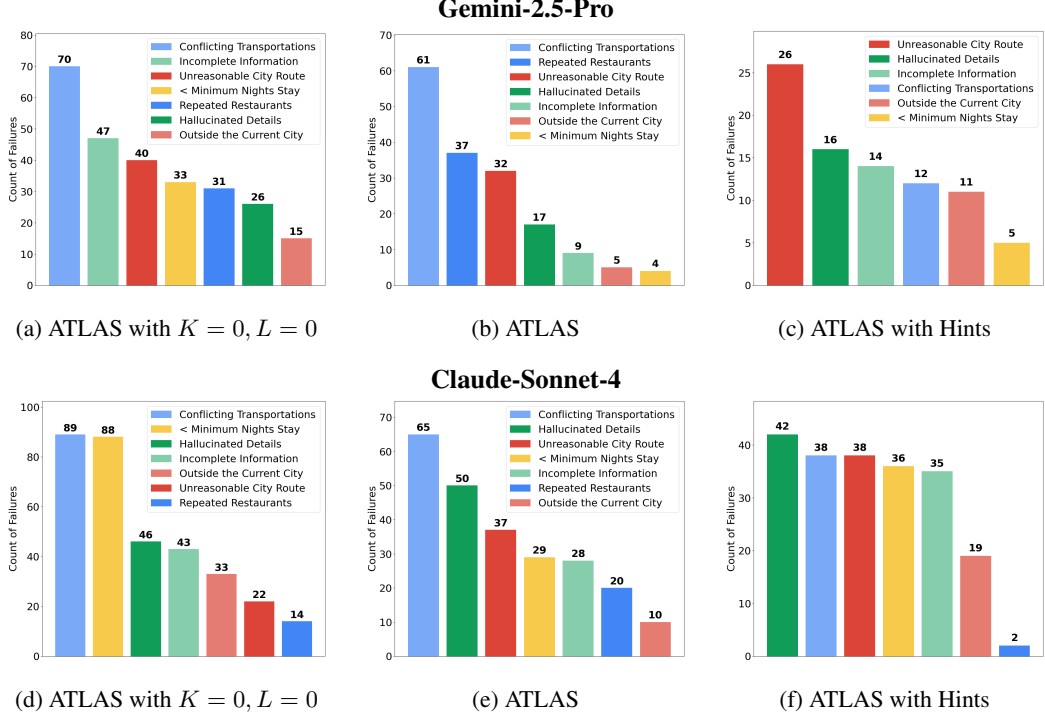

Figure 7: **Breakdown of commonsense constraint failures.** The figure illustrates the progressive reduction of errors from the (a) ATLAS without any check interactions or interleaved search (*i.e.,* ReAct), to (b) ATLAS with $K = 3, L = 10$, and finally to (c) ATLAS supplied with additional benchmark-specific constraints, following Zhang et al. (2025). Results are on the TravelPlanner validation set using Gemini-2.5-pro (top) or Claude-Sonnet-4 (bottom).

**Multi-Turn Travel Planning with Live Search.** We analyze the failure modes of each method in the multi-turn live search setting, observing how they address failures from the previous turn. As shown in Figure 8, simple constraint violations like repeated restaurant choices or conflicting transportation (light blue and blue bars) are easily resolved by all methods, since live search provides a virtually unlimited pool of alternatives.

However, more complex failures reveal key differences. The monolithic agent consistently fails to actually conduct the search to collect grounded context information rather than solely relying on its internal knowledge (see green bars in Figure 8a). While ReAct uses search to present the searched context information, it still hallucinates the details in its itinerary, a problem not effectively resolved even with explicit feedback (purple bars in Figure 8b). In contrast, ATLAS successfully grounds its plans in the search results and adheres to dynamic constraints from user feedback throughout turns.

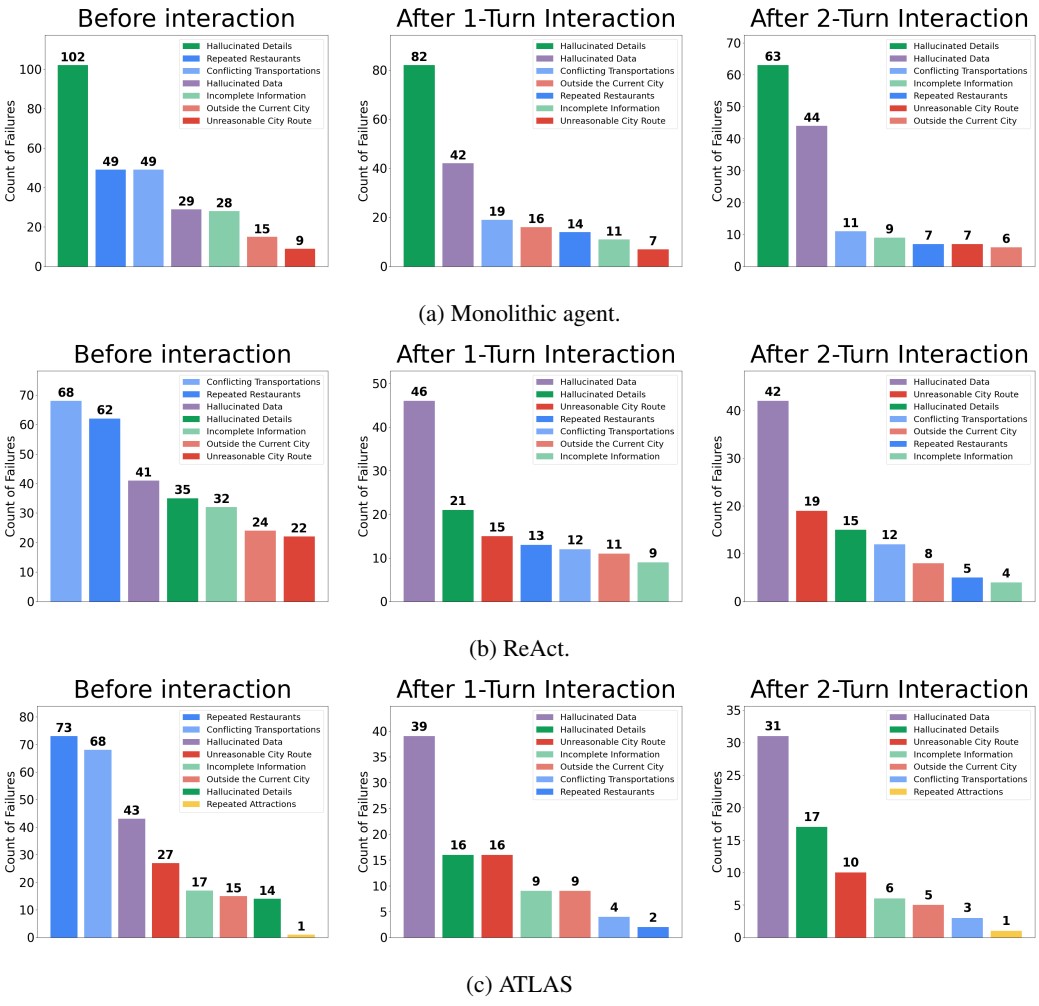

Figure 8: **Breakdown of commonsense constraint failure types on live travel planning.**

## D.6 ABLATIONS ON TRAVEL DAYS AND LEVELS

Table 9: **Ablation results on travel days.** We compare ATLAS to baselines on the TravelPlanner validation set (60 instances per day subset).

| Base Model | Subset | Method | Delivery ↑ | Commonsense ↑ | | Hard Constraint ↑ | | Final Pass ↑ |
|---|---|---|---|---|---|---|---|---|
| | | | | Micro | Macro | Micro | Macro | |
| Gemini-2.5-Pro | 3 day | ReAct | 100.00 | 94.38 | 65.00 | 67.86 | 51.67 | 45.00 |
| | | ReAct+Reflexion | 100.00 | 93.12 | 66.67 | 77.86 | 68.33 | 53.33 |
| | | ReAct+EvoAgent | 100.00 | 92.08 | 55.00 | 67.86 | 46.67 | 30.00 |
| | | PMC | 100.00 | 96.04 | 71.67 | 82.86 | 70.00 | 51.67 |
| | | ATLAS | **100.00** | **97.92** | **83.33** | **92.86** | **86.67** | **75.00** |
| | 5 day | ReAct | 100.00 | 75.83 | 16.67 | 49.29 | 40.00 | 11.67 |
| | | ReAct+Reflexion | 100.00 | 74.58 | 15.00 | 49.29 | 46.67 | 13.33 |
| | | ReAct+EvoAgent | 100.00 | 71.88 | 10.00 | 45.71 | 31.67 | 3.33 |
| | | PMC | 100.00 | 69.58 | 11.67 | 25.71 | 25.00 | 11.67 |
| | | ATLAS | **100.00** | **85.42** | **33.33** | **77.86** | **70.00** | **31.67** |
| | 7 day | ReAct | 98.33 | 73.54 | 15.00 | 51.43 | 48.33 | 11.67 |
| | | ReAct+Reflexion | 100.00 | 69.58 | 0.00 | 50.71 | 35.00 | 0.00 |
| | | ReAct+EvoAgent | 100.00 | 70.21 | 6.67 | 60.00 | 43.33 | 3.33 |
| | | PMC | 100.00 | 70.42 | 8.33 | 21.43 | 16.67 | 6.67 |
| | | ATLAS | **100.00** | **82.29** | **28.33** | **77.14** | **66.67** | **26.67** |
| Claude-Sonnet-4 | 3 day | ReAct | 100.00 | 88.96 | 45.00 | 68.57 | 50.00 | 28.33 |
| | | ReAct+Reflexion | 100.00 | 89.17 | 36.67 | 64.29 | 40.00 | 20.00 |
| | | ReAct+EvoAgent | 100.00 | 86.67 | 28.33 | 43.57 | 31.67 | 18.33 |
| | | PMC | 98.33 | 90.00 | 46.67 | 67.86 | 50.00 | 30.00 |
| | | ATLAS | **100.00** | **95.62** | **73.33** | **76.43** | **56.67** | **46.67** |
| | 5 day | ReAct | 100.00 | 73.12 | 6.67 | 44.29 | 35.00 | 3.33 |
| | | ReAct+Reflexion | 98.33 | 71.25 | 15.00 | 41.43 | 25.00 | 8.33 |
| | | ReAct+EvoAgent | 98.33 | 57.71 | 1.67 | 8.57 | 6.67 | 0.00 |
| | | PMC | 93.33 | 70.42 | 11.67 | 19.29 | 15.00 | 6.67 |
| | | ATLAS | **100.00** | **80.21** | **21.67** | **44.29** | **28.33** | **11.67** |
| | 7 day | ReAct | 100.00 | 68.12 | 5.00 | 41.43 | 30.00 | 3.33 |
| | | ReAct+Reflexion | 100.00 | 63.96 | 3.33 | 30.71 | 20.00 | 1.67 |
| | | ReAct+EvoAgent | 98.33 | 56.67 | 0.00 | 5.00 | 3.33 | 0.00 |
| | | PMC | 98.33 | 67.92 | 6.67 | 31.43 | 26.67 | 6.67 |
| | | ATLAS | **100.00** | **74.38** | **18.33** | **48.57** | **31.67** | **11.67** |

Interestingly, in Table 10, we observe that ATLAS performs better on the "hard" set than the "medium" set. We investigated this phenomenon and found that it is consistent with the inherent characteristics of the TravelPlanner benchmark, rather than an anomaly specific to ATLAS. For example, this trend is observable in Table 4 of the original TravelPlanner paper (Xie et al., 2024), where pass rates for "medium" are often lower than "hard" for both two-stage and sole-planning setups. Similarly, Table 3 of the PMC baseline (Zhang et al., 2025) reports a final pass rate for "medium" that is ∼8% lower than "hard". In Table 10, the baseline ReAct also exhibits this pattern (Medium: 11.67% vs. Hard: 21.67% final pass rate).

One possible hypothesis to explain this counter-intuitive observation is the trade-off between constraint complexity and domain set size (search space) in the constraint satisfaction problem.

- Easy to medium or hard: The increase in constraint complexity (*e.g.,* adding constraints beyond just budget) is the dominant factor, making the task strictly harder.

- Medium to hard: While "hard" tasks impose more constraints (adding transportation preferences on top of "medium" constraints), these specific constraints act as strong filters that prune the domain of applicable search results significantly. This reduces the number of *distractor* options for the planning agent, making the correct assignment of values for variables clearer.

Indeed, in Table 4 of Xie et al. (2024), as transportation preference constraints are added from medium to hard, there is a substantial increase in pass rates for transportation-relevant criteria *e.g.,*

"Non-conflicting Transportation" rises from 73.3% to 83.1% in two-stage, and 56.5% to 87.5% in sole-planning settings.

Table 10: **Ablation results on task difficulty levels.** We compare ATLAS to baselines on the TravelPlanner validation set (60 instances per level subset).

| Base Model | Subset | Method | Delivery ↑ | Commonsense ↑ | | Hard Constraint ↑ | | Final Pass ↑ |
|---|---|---|---|---|---|---|---|---|
| | | | | Micro | Macro | Micro | Macro | |
| Gemini-2.5-Pro | easy | ReAct | 100.00 | 84.17 | 35.00 | 73.33 | 73.33 | 35.00 |
| | | ReAct+Reflexion | 100.00 | 78.96 | 30.00 | 61.67 | 61.67 | 28.33 |
| | | ReAct+EvoAgent | 100.00 | 76.04 | 16.67 | 55.00 | 55.00 | 15.00 |
| | | PMC | 100.00 | 82.08 | 38.33 | 46.67 | 46.67 | 35.00 |
| | | ATLAS | **100.00** | **87.50** | **45.00** | **83.33** | **83.33** | **43.33** |
| | medium | ReAct | 98.33 | 74.38 | 23.33 | 45.00 | 31.67 | 11.67 |
| | | ReAct+Reflexion | 100.00 | 73.54 | 21.67 | 50.83 | 43.33 | 18.33 |
| | | ReAct+EvoAgent | 100.00 | 72.71 | 18.33 | 41.67 | 33.33 | 11.67 |
| | | PMC | 100.00 | 72.50 | 25.00 | 37.50 | 33.33 | 18.33 |
| | | ATLAS | **100.00** | **83.54** | **38.33** | **77.50** | **70.00** | **35.00** |
| | hard | ReAct | 100.00 | 85.21 | 38.33 | 57.50 | 35.00 | 21.67 |
| | | ReAct+Reflexion | 100.00 | 84.79 | 30.00 | 62.92 | 45.00 | 20.00 |
| | | ReAct+EvoAgent | 100.00 | 85.42 | 36.67 | 66.67 | 33.33 | 10.00 |
| | | PMC | 100.00 | 81.46 | 28.33 | 45.42 | 31.67 | 16.67 |
| | | ATLAS | **100.00** | **94.58** | **61.67** | **85.00** | **70.00** | **55.00** |
| Claude-Sonnet-4 | easy | ReAct | 100.00 | 72.50 | 13.33 | 50.00 | 50.00 | 11.67 |
| | | ReAct+Reflexion | 98.33 | 69.17 | 18.33 | 38.33 | 38.33 | 16.67 |
| | | ReAct+EvoAgent | 98.33 | 64.38 | 5.00 | 16.67 | 16.67 | 5.00 |
| | | PMC | 100.00 | 74.38 | 13.33 | 36.67 | 36.67 | 13.33 |
| | | ATLAS | **100.00** | **81.25** | **33.33** | **53.33** | **53.33** | **30.00** |
| | medium | ReAct | 100.00 | 73.54 | 18.33 | 44.17 | 35.00 | 11.67 |
| | | ReAct+Reflexion | 100.00 | 70.21 | 15.00 | 30.00 | 21.67 | 6.67 |
| | | ReAct+EvoAgent | 98.33 | 63.57 | 13.33 | 20.00 | 16.67 | 10.00 |
| | | PMC | 93.33 | 73.12 | 21.67 | 31.67 | 28.33 | 16.67 |
| | | ATLAS | **100.00** | **79.58** | **30.00** | **46.67** | **31.67** | **18.33** |
| | hard | ReAct | 100.00 | 84.17 | 25.00 | 55.42 | 30.00 | 11.67 |
| | | ReAct+Reflexion | 100.00 | 85.00 | 21.67 | 55.00 | 25.00 | 6.67 |
| | | ReAct+EvoAgent | 100.00 | 72.92 | 11.67 | 19.17 | 8.33 | 3.33 |
| | | PMC | 96.67 | 80.83 | 30.00 | 44.17 | 26.67 | 13.33 |
| | | ATLAS | **100.00** | **89.38** | **50.00** | **62.08** | **31.67** | **21.67** |

## D.7 Cost Analysis

To assess the real-world applicability of ATLAS, we conduct a detailed cost analysis of the framework. As shown in Figure 9, we measure the runtime for each agent on the TravelPlanner benchmark using Gemini-2.5-Pro as the base model (refer to Figure 10 for input/output token counts). For a three-day plan, the median runtime over 60 instances is approximately 6 minutes, and 15 minutes for 5- and 7-day plans. This demonstrates that ATLAS can resolve most planning requests within a reasonable time frame.

The analysis shows that the Planner agent is the most resource-intensive component, which is an inherent aspect of any complex planning task. In contrast, our constraint-related agents (*i.e.,* Checker and Constraint Manager) add minimal overhead relative to the significant performance gains they provide. Unsurprisingly, when live search is enabled, the Search agent becomes the primary driver of runtime, eclipsing the Planner (in Figure 9b). We also report the input and output token costs to each agent of ATLAS in Figure 10.

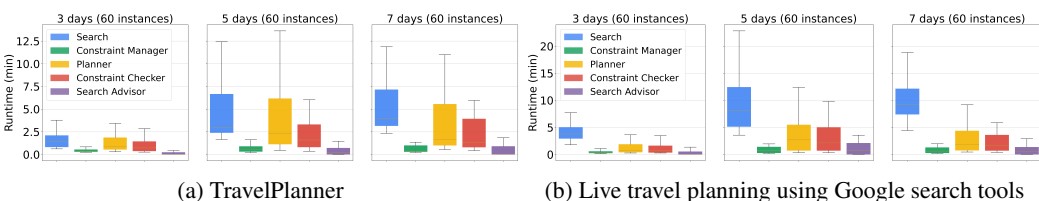

(a) TravelPlanner    (b) Live travel planning using Google search tools

Figure 9: **Wall clock runtime of ATLAS with Gemini-2.5-pro.**

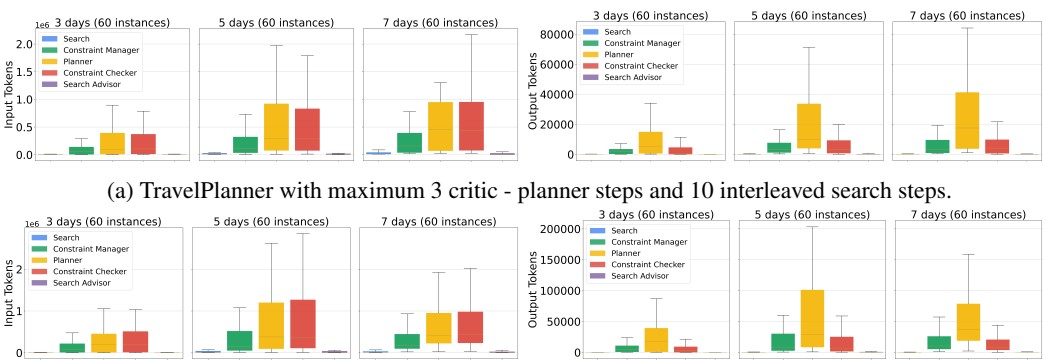

(a) TravelPlanner with maximum 3 critic - planner steps and 10 interleaved search steps.

(b) LivePlanner using Google search tools with maximum 3 critic - planner steps and 5 interleaved search steps.

Figure 10: **Token counts for ATLAS .** Total number of input (left) and output (right) tokens per each module of our framework.

Table 11 shows the cost analysis on all method. Multi-agentic approaches indeed take longer and requires more token costs, but when comparing the median, we observe that ATLAS does not take significantly more costs than other multi-agent baselines (*i.e.,* search-augmented EvoAgent or PMC) for the significant improvement it brings to the performance.

Table 11: **Cost analysis of ATLAS and baselines.** We compare the total wall clock runtime and sum of all output tokens. Results are on the TravelPlanner validation set using Gemini-2.5-pro. We report the (25th, 50th (median), 75th) percentiles.

| Method | 3 day | | 5 day | | 7 day | |
|---|---|---|---|---|---|---|
| | Runtime (min) | # Token (K) | Runtime (min) | # Token (K) | Runtime (min) | # Token (K) |
| ReAct | (1.20, 1.41, 1.74) | (0.33, 1.07, 17.05) | (2.42, 2.79, 3.28) | (0.71, 1.38, 2.09) | (3.49, 3.83, 4.49) | (1.03, 1.78, 2.26) |
| ReAct+Reflexion | (1.52, 2.29, 3.52) | (0.62, 0.84, 1.09) | (2.46, 4.37, 6.15) | (1.27, 1.55, 2.35) | (2.96, 5.41, 8.64) | (1.78, 2.01, 2.56) |
| ReAct+EvoAgent | (3.10, 3.36, 3.80) | (6.51, 7.13, 7.94) | (6.26, 7.10, 9.01) | (15.87, 19.36, 21.81) | (7.92, 8.76, 14.20) | (20.17, 22.90, 26.08) |
| PMC | (2.57, 2.85, 3.10) | (5.90, 6.69, 7.10) | (5.77, 7.19, 10.31) | (11.44, 12.72, 14.12) | (7.00, 8.35, 12.27) | (15.05, 16.56, 19.13) |
| ATLAS (ours) | (2.42, 4.07, 6.49) | (1.49, 9.52, 20.52) | (4.80, 8.26, 26.22) | (5.53, 19.29, 63.03) | (5.47, 7.97, 19.88) | (5.27, 24.07, 43.86) |

# E EXAMPLE OUTPUTS OF ATLAS

## E.1 END-TO-END EXAMPLES

### E.1.1 TRAVELPLANNER

**Example 1**

Query: Please create a travel plan for me where I'll be departing
    from Washington and heading to Myrtle Beach for a 3-day trip
    from March 13th to March 15th, 2022. Can you help me keep this
    journey within a budget of $1,400?

Output from Constraint Manager

1. The outbound flight from Washington to Myrtle Beach on March 13th
    , 2022 must be either flight F3792603 or F3927581.
2. The return flight from Myrtle Beach to Washington on March 15th,
    2022 must be either flight F3791200 or F3926434.
3. The chosen accommodation in Myrtle Beach must be selected from
    the provided list and have a minimum stay requirement of 2
    nights or less.
4. The house rules for the selected accommodation must be adhered to
    .
5. All restaurants visited in Myrtle Beach must be chosen from the
    provided list.
6. All attractions visited in Myrtle Beach must be chosen from the
    provided list.
7. The trip must start in Washington and return to Washington.
8. The destination city is Myrtle Beach.
9. The trip must take place from March 13th, 2022 to March 15th,
    2022.
10. The total cost of the trip must not exceed $1,400."

Output from Planner

Day 1:\nCurrent City: from Washington to Myrtle Beach\
    nTransportation: Flight Number: F3927581, from Washington to
    Myrtle Beach, Departure Time: 11:03, Arrival Time: 13:31\
    nBreakfast: -\nAttraction: Myrtle Beach Boardwalk and Promenade,
     Myrtle Beach\nLunch: First Eat, Myrtle Beach\nDinner: Catfish
    Charlie's, Myrtle Beach\nAccommodation: Yellow submarine, Myrtle
     Beach\n\nDay 2:\nCurrent City: Myrtle Beach\nTransportation: -\
    nBreakfast: d' Curry House, Myrtle Beach\nAttraction: Ripley's
    Aquarium of Myrtle Beach, Myrtle Beach;SkyWheel Myrtle Beach,
    Myrtle Beach.\nLunch: La Pino'z Pizza, Myrtle Beach\nDinner:
    Nizam's Kathi Kabab, Myrtle Beach\nAccommodation: Yellow
    submarine, Myrtle Beach\n\nDay 3:\nCurrent City: from Myrtle
    Beach to Washington\nTransportation: Flight Number: F3791200,
    from Myrtle Beach to Washington, Departure Time: 11:36, Arrival
    Time: 13:06\nBreakfast: Turning Point Fast Food, Myrtle Beach\
    nAttraction: Myrtle Beach State Park, Myrtle Beach.\nLunch: -\
    nDinner: -\nAccommodation: -

Output from Checker

Decision: valid

Output from Search Advisor

–

Final delivered plan

{

```
    "day": 1,
    "current_city": "from Washington to Myrtle Beach",
    "transportation": "Flight Number: F3927581, from Washington to
        Myrtle Beach, Departure Time: 11:03, Arrival Time: 13:31",
    "breakfast": "-",
    "attraction": "Myrtle Beach Boardwalk and Promenade, Myrtle
        Beach;",
    "lunch": "First Eat, Myrtle Beach",
    "dinner": "Catfish Charlie's, Myrtle Beach",
    "accommodation": "Yellow submarine, Myrtle Beach"
},
{
    "day": 2,
    "current_city": "Myrtle Beach",
    "transportation": "-",
    "breakfast": "d' Curry House, Myrtle Beach",
    "attraction": "Ripley's Aquarium of Myrtle Beach, Myrtle Beach;
        SkyWheel Myrtle Beach, Myrtle Beach;",
    "lunch": "La Pino'z Pizza, Myrtle Beach",
    "dinner": "Nizam's Kathi Kabab, Myrtle Beach",
    "accommodation": "Yellow submarine, Myrtle Beach"
},
{
    "day": 3,
    "current_city": "from Myrtle Beach to Washington",
    "transportation": "Flight Number: F3791200, from Myrtle Beach to
        Washington, Departure Time: 11:36, Arrival Time: 13:06",
    "breakfast": "Turning Point Fast Food, Myrtle Beach",
    "attraction": "Myrtle Beach State Park, Myrtle Beach;",
    "lunch": "-",
    "dinner": "-",
    "accommodation": "-"
}
```

**Example 2 where interleaved search easily addresses the information gap**

```
Query: Can you assist with a travel plan for one person departing
    from Pittsburgh to Baltimore for 3 days, from March 4th to March
    6th, 2022, with a maximum budget of $1,200?
```

Output from Search Agent after initial search step ($L = 0$)

```
{
    "index": 0,
    "Short Description": "Driving from Pittsburgh to Baltimore",
    "Content": "self-driving, from Pittsburgh to Baltimore, duration
        : 3 hours 58 mins, distance: 400 km, cost: 20"
},
{
    "index": 1,
    "Short Description": "Accommodation in Baltimore",
    "Content": ...
},
{
    "index": 2,
    "Short Description": "Attractions in Baltimore",
    "Content": ...
},
{
    "index": 3,
    "Short Description": "Restaurants in Baltimore",
    "Content": ...
}
```

Output from Constraint Manager at $L = 0$

```
1. The minimum stay for accommodation in Baltimore must be 2 nights
    or less.
2. The round trip transportation cost between Pittsburgh and
    Baltimore is $40.
3. The trip must start from Pittsburgh and return to Pittsburgh.
4. The destination city is Baltimore.
5. The trip is for one person.
6. The trip must be for 3 days, from March 4th, 2022, to March 6th,
    2022.
7. The total cost of the trip must not exceed $1,200.",
```

Output from Planner at $L = 0, K = 0$

```
Day 1:\nCurrent City: from Pittsburgh to Baltimore\nTransportation:
    self-driving, from Pittsburgh to Baltimore\nBreakfast: -\
    nAttraction: Inner Harbor, Baltimore\nLunch: Mr. Dunderbak's
    Biergarten and Marketplatz, Baltimore\nDinner: Los Pablos,
    Baltimore\nAccommodation: Contemporary Home Away from Home,
    Entire house, Baltimore\n\nDay 2:\nCurrent City: Baltimore\
    nTransportation: -\nBreakfast: Amalfi, Baltimore\nAttraction:
    National Aquarium, Baltimore;Fort McHenry National Monument and
    Historic Shrine, Baltimore.\nLunch: Salt, Baltimore\nDinner: The
     Thai Bowl, Baltimore\nAccommodation: Contemporary Home Away
    from Home, Entire house, Baltimore\n\nDay 3:\nCurrent City: from
     Baltimore to Pittsburgh\nTransportation: self-driving, from
    Baltimore to Pittsburgh\nBreakfast: Tibb's Frankie, Baltimore\
    nAttraction: The Walters Art Museum, Baltimore.\nLunch:
    RollsKing, Baltimore\nDinner: -\nAccommodation: -\n
```

Output from Checker at $L = 0, K = 1$

```
Decision 1: unsatisfiable
```

```
Feedback 1: The plan is invalid because the transportation from
    Baltimore to Pittsburgh on Day 3 is not derived from the
    provided context information. This failure is due to the lack of
     information about the return trip in the context, which is
    necessary to fulfill the constraint of returning to the origin
    city."
```

Output from Search Advisor at $L = 0$

```
Suggested actions:
FlightSearch[Pittsburgh, Baltimore, 2022-03-04]
NotebookWrite[Flights from Pittsburgh to Baltimore on 2022-03-04]
FlightSearch[Baltimore, Pittsburgh, 2022-03-06]
NotebookWrite[Flights from Baltimore to Pittsburgh on 2022-03-06]"
```

Output from Search Agent after the first interleaved search step ($L = 1$)

```
{
    "index": 0,
    "Short Description": "Driving from Pittsburgh to Baltimore",
    "Content": "self-driving, from Pittsburgh to Baltimore, duration
        : 3 hours 58 mins, distance: 400 km, cost: 20"
},
{
    "index": 1,
    "Short Description": "Accommodation in Baltimore",
    "Content": ...
},
{
    "index": 2,
    "Short Description": "Attractions in Baltimore",
    "Content": ...
},
{
    "index": 3,
    "Short Description": "Restaurants in Baltimore",
    "Content": ...
},
{
    "index": 4,
    "Short Description": "Flights from Pittsburgh to Baltimore on
        2022-03-04",
    "Content": ...
},
{
    "index": 5,
    "Short Description": "Flights from Baltimore to Pittsburgh on
        2022-03-06",
    "Content":  ...
}
```

Output from Constraint Manager at $L = 1$

```
1. The accommodation in Baltimore must have a minimum stay
    requirement of 2 nights or less.
2. The trip must start in Pittsburgh and end in Pittsburgh.
3. The trip must be from March 4th, 2022, to March 6th, 2022.
4. The trip is for one person.
5. The total cost must not exceed $1,200.
6. The destination city is Baltimore."
```

Output from Planner at $L = 1, K = 0$

```
Day 1:\nCurrent City: from Pittsburgh to Baltimore\nTransportation:
    Flight Number: F3969954, from Pittsburgh to Baltimore, Departure
     Time: 16:51, Arrival Time: 17:49\nBreakfast: -\nAttraction: -\
```

```
    nLunch: -\nDinner: Mr. Dunderbak's Biergarten and Marketplatz,
    Baltimore\nAccommodation: Contemporary Home Away from Home,
    Entire house, Baltimore\n\nDay 2:\nCurrent City: Baltimore\
    nTransportation: -\nBreakfast: Amalfi, Baltimore\nAttraction:
    National Aquarium, Baltimore;Fort McHenry National Monument and
    Historic Shrine, Baltimore.\nLunch: Los Pablos, Baltimore\
    nDinner: Farzi Cafe, Baltimore\nAccommodation: Contemporary Home
     Away from Home, Entire house, Baltimore\n\nDay 3:\nCurrent City
    : from Baltimore to Pittsburgh\nTransportation: Flight Number:
    F3994096, from Baltimore to Pittsburgh, Departure Time: 21:45,
    Arrival Time: 22:44\nBreakfast: 28 Capri Italy, Baltimore\
    nAttraction: The Walters Art Museum, Baltimore.\nLunch: Tresind
    - Nassima Royal Hotel, Baltimore\nDinner: The Manhattan Fish
    Market, Baltimore\nAccommodation: -
```

Output from Checker at $L = 1, K = 1$

```
Decision: valid
```

Final delivered plan

```
{
    "day": 1,
    "current_city": "from Pittsburgh to Baltimore",
    "transportation": "Flight Number: F3969954, from Pittsburgh to
        Baltimore, Departure Time: 16:51, Arrival Time: 17:49",
    "breakfast": "-",
    "attraction": "-",
    "lunch": "-",
    "dinner": "Mr. Dunderbak's Biergarten and Marketplatz, Baltimore
        ",
    "accommodation": "Contemporary Home Away from Home, Baltimore"
},
{
    "day": 2,
    "current_city": "Baltimore",
    "transportation": "-",
    "breakfast": "Amalfi, Baltimore",
    "attraction": "National Aquarium, Baltimore;Fort McHenry
        National Monument and Historic Shrine, Baltimore;",
    "lunch": "Los Pablos, Baltimore",
    "dinner": "Farzi Cafe, Baltimore",
    "accommodation": "Contemporary Home Away from Home, Baltimore"
},
{
    "day": 3,
    "current_city": "from Baltimore to Pittsburgh",
    "transportation": "Flight Number: F3994096, from Baltimore to
        Pittsburgh, Departure Time: 21:45, Arrival Time: 22:44",
    "breakfast": "28 Capri Italy, Baltimore",
    "attraction": "The Walters Art Museum, Baltimore;",
    "lunch": "Tresind - Nassima Royal Hotel, Baltimore",
    "dinner": "The Manhattan Fish Market, Baltimore",
    "accommodation": "-"
}
```

**Example 3 where plan is extensively revised**

```
Query: Can you create a travel plan for a group of 5 departing from
    Charlotte heading to Hilton Head, to be carried out over 3 days,
     from March 26th to March 28th, 2022? The budget for this trip
    is capped at $7,000. We have a preference for Italian and French
     cuisines during our trip.
```

Output from Search Agent after initial search step ($L = 0$)

```
{
    "index": 0,
    "Short Description": "Transportation from Charlotte to Hilton
        Head",
    "Content": "self-driving, from Charlotte to Hilton Head,
        duration: 3 hours 49 mins, distance: 398 km, cost: 19"
},
{
    "index": 1,
    "Short Description": "Accommodation in Hilton Head",
    "Content": "                                            NAME
        price       room type                        house_rules
        minimum nights  maximum occupancy  review rate number
                city
    Williamsburg Home Away From Home!  164.0 Entire home/apt
                                    No pets              4.0
                     3              3.0 Hilton Head
    COZY Room @Williamsburg (10 mins to Manhattan)  605.0    Private
        room                     No visitors              2.0
                     1              4.0 Hilton Head
    ...
},
{
    "index": 2,
    "Short Description": "Restaurants in Hilton Head",
    "Content": "                            Name  Average Cost
                                    Cuisines  Aggregate
        Rating       City
    Taste Of China           91            Tea, Seafood, Bakery,
        Fast Food             4.0 Hilton Head
    K Raga's          71          Tea, BBQ, Mediterranean,
        Desserts             2.9 Hilton Head
    New Town Pastry Shop - Park Plaza           51
                            Tea, Cafe, Pizza, BBQ
        3.2 Hilton Head
    ...
},
{
    "index": 3,
    "Short Description": "Attractions in Hilton Head",
    "Content": "                            Name  Latitude
        Longitude
                                                    Address
                Phone

        Website       City
    Coastal Discovery Museum 32.211485 -80.744025        70 Honey
        Horn Dr, Hilton Head Island, SC 29926, USA (843) 689-6767

        http://www.coastaldiscovery.org/ Hilton Head
    ...
}
```

Output from Constraint Manager at $L = 0$

```
1. The minimum stay for accommodation must be less than or equal to
    2 nights.
2. The plan must include at least one restaurant that serves Italian
    cuisine and one that serves French cuisine.
3. The mode of transportation must be self-driving.
4. The trip must start in Charlotte and end in Charlotte.
5. The trip must be for 5 people.
6. The trip must take place from March 26th, 2022 to March 28th,
    2022.
7. The total cost of the trip must not exceed $7,000.
```

Output from Planner at $L = 0, K = 0$

```
Day 1:\nCurrent City: from Charlotte to Hilton Head\nTransportation:
    self-driving, from Charlotte to Hilton Head\nBreakfast: -\
    nAttraction: Harbour Town Lighthouse, Hilton Head\nLunch:
    Wrapster, Hilton Head\nDinner: Dhaba Ambarsariya, Hilton Head\
    nAccommodation: Hip, Vibrant, COLORFUL Downtown Manhattan 1 Bed,
     Hilton Head\n\nDay 2:\nCurrent City: Hilton Head\
    nTransportation: -\nBreakfast: Cafe Coffee Day, Hilton Head\
    nAttraction: Coastal Discovery Museum, Hilton Head; Coligny
    Beach Park, Hilton Head.\nLunch: Mr. Brown, Hilton Head\nDinner:
     Sikkim Fast Food, Hilton Head\nAccommodation: Hip, Vibrant,
    COLORFUL Downtown Manhattan 1 Bed, Hilton Head\n\nDay 3:\
    nCurrent City: from Hilton Head to Charlotte\nTransportation:
    self-driving, from Hilton Head to Charlotte\nBreakfast:
    Connoisseur, Hilton Head\nAttraction: Sea Pines Forest Preserve,
     Hilton Head.\nLunch: MR.D - Deliciousness Delivered, Hilton
    Head\nDinner: -\nAccommodation: -
```

Output from Checker at $L = 0, K = 1$

```
Decision: unsatisfiable
Feedback: The plan is invalid because it includes transportation
    from Hilton Head to Charlotte on Day 3, which is not detailed in
     the provided context information. This failure is due to the
    context information being incomplete, as it lacks the necessary
    details for the return trip, which is required by the constraint
     that the trip must end in Charlotte.
```

Output from Search Advisor at $L = 1$

```
FlightSearch[Charlotte, Hilton Head, 2022-03-26]
NotebookWrite[Flights from Charlotte to Hilton Head on 2022-03-26]
FlightSearch[Hilton Head, Charlotte, 2022-03-28]
NotebookWrite[Flights from Hilton Head to Charlotte on 2022-03-28]
```

Output from Search Agent after first interleaved search step ($L = 1$)

```
{
    "index": 0,
    "Short Description": "Transportation from Charlotte to Hilton
        Head",
    "Content": "self-driving, from Charlotte to Hilton Head,
        duration: 3 hours 49 mins, distance: 398 km, cost: 19"
},
{
    "index": 1,
    "Short Description": "Accommodation in Hilton Head",
    "Content": "                                          NAME
        price        room type                    house_rules
        minimum nights  maximum occupancy  review rate number
                city
```

```
        Williamsburg Home Away From Home!  164.0 Entire home/apt
                                   No pets                    4.0
                          3               3.0 Hilton Head
        COZY Room @Williamsburg (10 mins to Manhattan) 605.0    Private
            room                        No visitors            2.0
                          1               4.0 Hilton Head
        ...
    },
    {
        "index": 2,
        "Short Description": "Restaurants in Hilton Head",
        "Content": "                                Name  Average Cost
                                        Cuisines  Aggregate
            Rating        City
        Taste Of China              91           Tea, Seafood, Bakery,
            Fast Food             4.0 Hilton Head
        K Raga's           71          Tea, BBQ, Mediterranean,
            Desserts             2.9 Hilton Head
        New Town Pastry Shop - Park Plaza          51
                               Tea, Cafe, Pizza, BBQ
            3.2 Hilton Head
        ...
    },
    {
        "index": 3,
        "Short Description": "Attractions in Hilton Head",
        "Content": "                               Name  Latitude
            Longitude
                                                         Address
                    Phone
            Website        City
        Coastal Discovery Museum 32.211485 -80.744025          70 Honey
            Horn Dr, Hilton Head Island, SC 29926, USA (843) 689-6767

            http://www.coastaldiscovery.org/ Hilton Head
        ...
    },
    {
        "index": 4,
        "Short Description": "Flights from Charlotte to Hilton Head on
            2022-03-26",
        "Content": "Flight Number  Price DepTime ArrTime
            ActualElapsedTime FlightDate OriginCityName DestCityName
            Distance
        F4055090     92   18:07   19:27 1 hours 20 minutes 2022-03-26
                Charlotte  Hilton Head    207.0
        F4059890     95   13:01   14:22 1 hours 21 minutes 2022-03-26
                Charlotte  Hilton Head    207.0"
    },
    {
        "index": 5,
        "Short Description": "Flights from Hilton Head to Charlotte on
            2022-03-28",
        "Content": "Flight Number  Price DepTime ArrTime
            ActualElapsedTime FlightDate OriginCityName DestCityName
            Distance     F4056985    46   20:07   21:31 1 hours 24
            minutes 2022-03-28    Hilton Head    Charlotte    207.0
        F4059921     45   14:52   16:04 1 hours 12 minutes 2022-03-28
                Hilton Head    Charlotte    207.0"
    }
```

Output from Constraint Manager at $L = 1$

```
   1. The selected accommodation in Hilton Head must have a minimum
      stay requirement of 2 nights or less.
   2. Meals must be chosen from the following restaurants that serve
      French or Italian cuisine: Sikkim Fast Food, Cafe Coffee Day,
      Dhaba Ambarsariya, Connoisseur, MR.D - Deliciousness Delivered,
      Mr. Brown, or Wrapster.
   3. The travel plan must use one of the two available modes of
      transport: self-driving or one of the specified flights.
   4. The trip must start in Charlotte and return to Charlotte.
   5. The trip must be for a group of 5 people.
   6. The trip must take place from March 26th, 2022, to March 28th,
      2022.
   7. The total cost of the trip must not exceed $7,000."
```

Output from Planner at $L = 1, K = 0$

```
Day 1:\nCurrent City: from Charlotte to Hilton Head\nTransportation:
    self-driving, from Charlotte to Hilton Head\nBreakfast: -\
   nAttraction: Coligny Beach Park, Hilton Head\nLunch: -\nDinner:
   Dhaba Ambarsariya, Hilton Head\nAccommodation: Hip, Vibrant,
   COLORFUL Downtown Manhattan 1 Bed, Hilton Head\n\nDay 2:\
   nCurrent City: Hilton Head\nTransportation: -\nBreakfast:
   Wrapster, Hilton Head\nAttraction: Harbour Town Lighthouse,
   Hilton Head; Coastal Discovery Museum, Hilton Head.\nLunch: Mr.
   Brown, Hilton Head\nDinner: Sikkim Fast Food, Hilton Head\
   nAccommodation: Hip, Vibrant, COLORFUL Downtown Manhattan 1 Bed,
    Hilton Head\n\nDay 3:\nCurrent City: from Hilton Head to
   Charlotte\nTransportation: self-driving, from Hilton Head to
   Charlotte\nBreakfast: Cafe Coffee Day, Hilton Head\nAttraction:
   Sea Pines Forest Preserve, Hilton Head.\nLunch: Connoisseur,
   Hilton Head\nDinner: -\nAccommodation: -
```

Output from Checker at $L = 1, K = 1$

```
Decision: invalid
Feedback:
1. The transportation for Day 3, 'self-driving, from Hilton Head to
    Charlotte', is not supported by the provided context information
    . The context only includes details for self-driving from
    Charlotte to Hilton Head.
```

Output from Planner at $L = 1, K = 1$

```
Day 1:\nCurrent City: from Charlotte to Hilton Head\nTransportation:
    self-driving, from Charlotte to Hilton Head\nBreakfast: -\
   nAttraction: Coligny Beach Park, Hilton Head\nLunch: -\nDinner:
   Dhaba Ambarsariya, Hilton Head\nAccommodation: Hip, Vibrant,
   COLORFUL Downtown Manhattan 1 Bed, Hilton Head\n\nDay 2:\
   nCurrent City: Hilton Head\nTransportation: -\nBreakfast:
   Wrapster, Hilton Head\nAttraction: Harbour Town Lighthouse,
   Hilton Head; Coastal Discovery Museum, Hilton Head.\nLunch: Mr.
   Brown, Hilton Head\nDinner: Sikkim Fast Food, Hilton Head\
   nAccommodation: Hip, Vibrant, COLORFUL Downtown Manhattan 1 Bed,
    Hilton Head\n\nDay 3:\nCurrent City: from Hilton Head to
   Charlotte\nTransportation: Flight Number: F4059921, from Hilton
   Head to Charlotte, Departure Time: 14:52, Arrival Time: 16:04\
   nBreakfast: Cafe Coffee Day, Hilton Head\nAttraction: Sea Pines
   Forest Preserve, Hilton Head.\nLunch: Connoisseur, Hilton Head\
   nDinner: -\nAccommodation: -
```

Output from Checker at $L = 1, K = 2$

```
Decision: invalid
Feedback:
1. The transportation plan is not logical. The travelers self-drive
    to Hilton Head on Day 1 but take a flight back to Charlotte on
```

```
    Day 3, which would result in their car being left behind in
    Hilton Head.
```

Output from Planner at $L = 1, K = 2$

```
Day 1:\nCurrent City: from Charlotte to Hilton Head\nTransportation:
    Flight Number: F4059890, from Charlotte to Hilton Head,
    Departure Time: 13:01, Arrival Time: 14:22\nBreakfast: -\
    nAttraction: Coligny Beach Park, Hilton Head\nLunch: -\nDinner:
    Dhaba Ambarsariya, Hilton Head\nAccommodation: Hip, Vibrant,
    COLORFUL Downtown Manhattan 1 Bed, Hilton Head\n\nDay 2:\
    nCurrent City: Hilton Head\nTransportation: -\nBreakfast:
    Wrapster, Hilton Head\nAttraction: Harbour Town Lighthouse,
    Hilton Head; Coastal Discovery Museum, Hilton Head.\nLunch: Mr.
    Brown, Hilton Head\nDinner: Sikkim Fast Food, Hilton Head\
    nAccommodation: Hip, Vibrant, COLORFUL Downtown Manhattan 1 Bed,
     Hilton Head\n\nDay 3:\nCurrent City: from Hilton Head to
    Charlotte\nTransportation: Flight Number: F4056985, from Hilton
    Head to Charlotte, Departure Time: 20:07, Arrival Time: 21:31\
    nBreakfast: Cafe Coffee Day, Hilton Head\nAttraction: Sea Pines
    Forest Preserve, Hilton Head.\nLunch: Connoisseur, Hilton Head\
    nDinner: -\nAccommodation: -\n
```

Output from Checker at $L = 1, K = 3$

```
Decision: valid
```

Final delivered plan

```
{
    "day": 1,
    "current_city": "from Charlotte to Hilton Head",
    "transportation": "Flight Number: F4059890, from Charlotte to
        Hilton Head, Departure Time: 13:01, Arrival Time: 14:22",
    "breakfast": "-",
    "attraction": "Coligny Beach Park, Hilton Head;",
    "lunch": "-",
    "dinner": "Dhaba Ambarsariya, Hilton Head",
    "accommodation": "Hip, Vibrant, COLORFUL Downtown Manhattan 1
        Bed, Hilton Head"
},
{
    "day": 2,
    "current_city": "Hilton Head",
    "transportation": "-",
    "breakfast": "Wrapster, Hilton Head",
    "attraction": "Harbour Town Lighthouse, Hilton Head;Coastal
        Discovery Museum, Hilton Head;",
    "lunch": "Mr. Brown, Hilton Head",
    "dinner": "Sikkim Fast Food, Hilton Head",
    "accommodation": "Hip, Vibrant, COLORFUL Downtown Manhattan 1
        Bed, Hilton Head"
},
{
    "day": 3,
    "current_city": "from Hilton Head to Charlotte",
    "transportation": "Flight Number: F4056985, from Hilton Head to
        Charlotte, Departure Time: 20:07, Arrival Time: 21:31",
    "breakfast": "Cafe Coffee Day, Hilton Head",
    "attraction": "Sea Pines Forest Preserve, Hilton Head;",
    "lunch": "Connoisseur, Hilton Head",
    "dinner": "-",
    "accommodation": "-"
}
```

### E.1.2 LIVE TRAVEL PLANNING WITH MULTI-TURN FEEDBACK

**First Turn**

```
Query: Could you help me arrange a 7-day solo travel itinerary from
    Kona to California with a budget of $5,800, intending to visit 3
     distinct cities in California from September 7th to September
    13th, 2025?
```

Output from Constraint Manager

```
1. The traveler must stay in Los Angeles from September 7th to
    September 9th, 2025.
2. The traveler must stay in San Francisco from September 9th to
    September 11th, 2025.
3. The traveler must stay in San Diego from September 11th to
    September 13th, 2025.
4. The three cities to be visited are Los Angeles, San Francisco,
    and San Diego.
5. The mode of transportation between cities in California must be
    self-driving.
6. The trip must start with a flight from Kona to Los Angeles on
    September 7th, 2025.
7. The trip must end with a flight from San Diego to Kona on
    September 13th, 2025.
8. The total trip duration is 7 days, from September 7th, 2025 to
    September 13th, 2025.
9. The total budget for the trip must not exceed $5,800.
10. The trip is for a solo traveler.
```

Final delivered plan

```
{
    "day": 1,
    "current_city": "from Kona to Los Angeles",
    "transportation": "Flight Number: HA, from Kona to Los Angeles,
        Departure Time: 15:19, Arrival Time: 23:51",
    "breakfast": "-",
    "attraction": "-",
    "lunch": "-",
    "dinner": "-",
    "accommodation": "Modern Homestay 2, Los Angeles"
},
{
    "day": 2,
    "current_city": "Los Angeles",
    "transportation": "-",
    "breakfast": "Humble Bird, Los Angeles",
    "attraction": "Griffith Observatory, Los Angeles;Hollywood Walk
        of Fame, Los Angeles;",
    "lunch": "Granville, Los Angeles",
    "dinner": "Chao Krung Thai, Los Angeles",
    "accommodation": "Modern Homestay 2, Los Angeles"
},
{
    "day": 3,
    "current_city": "from Los Angeles to San Francisco",
    "transportation": "self-driving, from Los Angeles to San
        Francisco",
    "breakfast": "La Pupusa Urban Eatery, Los Angeles",
    "attraction": "The Getty Center, Los Angeles;",
    "lunch": "-",
    "dinner": "Sweet Maple, San Francisco",
```

```
    "accommodation": "Travelodge by Wyndham San Francisco Airport
        North, San Francisco"
},
{
    "day": 4,
    "current_city": "San Francisco",
    "transportation": "-",
    "breakfast": "Sweet Maple, San Francisco",
    "attraction": "Golden Gate Bridge, San Francisco;Fisherman's
        Wharf, San Francisco;",
    "lunch": "Bar Crudo, San Francisco",
    "dinner": "Zuni Cafe, San Francisco",
    "accommodation": "Travelodge by Wyndham San Francisco Airport
        North, San Francisco"
},
{
    "day": 5,
    "current_city": "from San Francisco to San Diego",
    "transportation": "self-driving, from San Francisco to San Diego
        ",
    "breakfast": "Flour + Water, San Francisco",
    "attraction": "Alcatraz Island, San Francisco;",
    "lunch": "-",
    "dinner": "Oscar's Mexican Seafood, San Diego",
    "accommodation": "Quaint And Cozy One Bedroom Unit With Patio,
        San Diego"
},
{
    "day": 6,
    "current_city": "San Diego",
    "transportation": "-",
    "breakfast": "Las Cuatro Milpas, San Diego",
    "attraction": "Balboa Park, San Diego;USS Midway Museum, San
        Diego;",
    "lunch": "Mitch's Seafood, San Diego",
    "dinner": "Puesto, San Diego",
    "accommodation": "Quaint And Cozy One Bedroom Unit With Patio,
        San Diego"
},
{
    "day": 7,
    "current_city": "from San Diego to Kona",
    "transportation": "Flight Number: HA, from San Diego to Kona,
        Departure Time: 11:17, Arrival Time: 14:19",
    "breakfast": "Kindred, San Diego",
    "attraction": "-",
    "lunch": "-",
    "dinner": "-",
    "accommodation": "-"
}
```

**Second Turn**

```
Query: Could you help me arrange a 7-day solo travel itinerary from
    Kona to California with a budget of $5,800, intending to visit 3
    distinct cities in California from September 7th to September
    13th, 2025?

And this was the plan you generated previously.
Day 1:Current City: from Kona to Los AngelesTransportation: Flight
    Number: HA, from Kona to Los Angeles, Departure Time: 15:19,
    Arrival Time: 23:51Breakfast: -Attraction: -Lunch: -Dinner: -
    Accommodation: Modern Homestay 2, Los AngelesDay 2:Current City:
     Los AngelesTransportation: -Breakfast: Humble Bird, Los
    AngelesAttraction: Griffith Observatory, Los Angeles; Hollywood
    Walk of Fame, Los Angeles.Lunch: Granville, Los AngelesDinner:
    Chao Krung Thai, Los AngelesAccommodation: Modern Homestay 2,
    Los AngelesDay 3:Current City: from Los Angeles to San
    FranciscoTransportation: self-driving, from Los Angeles to San
    FranciscoBreakfast: La Pupusa Urban Eatery, Los
    AngelesAttraction: The Getty Center, Los Angeles.Lunch: -Dinner:
     Sweet Maple, San FranciscoAccommodation: Travelodge by Wyndham
    San Francisco Airport North, San FranciscoDay 4:Current City:
    San FranciscoTransportation: -Breakfast: Sweet Maple, San
    FranciscoAttraction: Golden Gate Bridge, San Francisco;
    Fisherman's Wharf, San Francisco.Lunch: Bar Crudo, San
    FranciscoDinner: Zuni Cafe, San FranciscoAccommodation:
    Travelodge by Wyndham San Francisco Airport North, San
    FranciscoDay 5:Current City: from San Francisco to San
    DiegoTransportation: self-driving, from San Francisco to San
    DiegoBreakfast: Flour + Water, San FranciscoAttraction: Alcatraz
     Island, San Francisco.Lunch: -Dinner: Oscar's Mexican Seafood,
    San DiegoAccommodation: Quaint And Cozy One Bedroom Unit With
    Patio, San DiegoDay 6:Current City: San DiegoTransportation: -
    Breakfast: Las Cuatro Milpas, San DiegoAttraction: Balboa Park,
    San Diego; USS Midway Museum, San Diego.Lunch: Mitch's Seafood,
    San DiegoDinner: Puesto, San DiegoAccommodation: Quaint And Cozy
     One Bedroom Unit With Patio, San DiegoDay 7:Current City: from
    San Diego to KonaTransportation: Flight Number: HA, from San
    Diego to Kona, Departure Time: 11:17, Arrival Time: 14:19
    Breakfast: Kindred, San DiegoAttraction: -Lunch: -Dinner: -
    Accommodation: -

Let me also tell you the major issues on this plan.
The restaurant in day 4 breakfast is repeated. I want to try various
     restaurants and attractions.
The transportation is conflicting. If I travel by air or taxi, I
    cannot do self-driving at any point of the trip. Conversely, if
    I'm able to use a self-driving car, I will not require a taxi or
     a flight.
The breakfast in day 2 is invalid in the sandbox.
The accommodation information in day 1 is hallucinated.
```

Output from Constraint Manager

```
1. The transportation method must be consistent: either use self-
    driving for all inter-city travel or use flights/taxis, but not
    a mix of both.
2. All selected restaurants and attractions must be unique and not
    repeated.
3. The restaurant 'Humble Bird' must not be included in the plan.
4. The accommodation 'Modern Homestay 2' must not be included in the
    plan.
```

```
  5. All flights, accommodations, restaurants, and attractions must be
       selected exclusively from the provided context information.
  6. The trip must start in Kona on September 7th, 2025 and end in
       Kona on September 13th, 2025.
  7. The itinerary must include visits to three distinct cities: Los
       Angeles, San Francisco, and San Diego.
  8. The total budget for the trip must not exceed $5,800.
  9. The trip is for a solo traveler.
  10. The trip must be a closed loop, starting from Kona and returning
       to Kona.
  11. Accommodation in Los Angeles must be booked from September 7th
       to September 9th, 2025.
  12. Accommodation in San Francisco must be booked from September 9th
       to September 11th, 2025.
  13. Accommodation in San Diego must be booked from September 11th to
       September 13th, 2025.
```

Output from Search Advisor

```
GoogleSearchFlight[Los Angeles, San Francisco, 2025-09-09]
NotebookWrite[Flights from Los Angeles to San Francisco on
    2025-09-09]
GoogleSearchFlight[San Francisco, San Diego, 2025-09-11]
NotebookWrite[Flights from San Francisco to San Diego on 2025-09-11]
```

Final delivered plan

```
{
    "day": 1,
    "current_city": "from Kona to Los Angeles",
    "transportation": "Flight Number: HA, from Kona to Los Angeles,
        Departure Time: 15:19, Arrival Time: 23:51",
    "breakfast": "-",
    "attraction": "-",
    "lunch": "-",
    "dinner": "-",
    "accommodation": "Modern Hostel, Los Angeles"
},
{
    "day": 2,
    "current_city": "Los Angeles",
    "transportation": "-",
    "breakfast": "La Pupusa Urban Eatery, Los Angeles",
    "attraction": "Griffith Observatory, Los Angeles;Hollywood Walk
        of Fame, Los Angeles;",
    "lunch": "Granville, Los Angeles",
    "dinner": "Chao Krung Thai, Los Angeles",
    "accommodation": "Modern Hostel, Los Angeles"
},
{
    "day": 3,
    "current_city": "from Los Angeles to San Francisco",
    "transportation": "Flight Number: HA, from Los Angeles to San
        Francisco, Departure Time: 22:34, Arrival Time: 23:59",
    "breakfast": "Holbox, Los Angeles",
    "attraction": "The Getty Center, Los Angeles;",
    "lunch": "Niku X, Los Angeles",
    "dinner": "The Brothers Sushi, Los Angeles",
    "accommodation": "Travelodge by Wyndham San Francisco Airport
        North, San Francisco"
},
{
    "day": 4,
    "current_city": "San Francisco",
    "transportation": "-",
```

```
        "breakfast": "Sweet Maple, San Francisco",
        "attraction": "Golden Gate Bridge, San Francisco;Fisherman's
            Wharf, San Francisco;",
        "lunch": "Bar Crudo, San Francisco",
        "dinner": "Zuni Cafe, San Francisco",
        "accommodation": "Travelodge by Wyndham San Francisco Airport
            North, San Francisco"
    },
    {
        "day": 5,
        "current_city": "from San Francisco to San Diego",
        "transportation": "Flight Number: Southwest, from San Francisco
            to San Diego, Departure Time: 11:00, Arrival Time: 12:35",
        "breakfast": "Flour + Water, San Francisco",
        "attraction": "Alcatraz Island, San Francisco;",
        "lunch": "-",
        "dinner": "Oscar's Mexican Seafood, San Diego",
        "accommodation": "Quaint And Cozy One Bedroom Unit With Patio,
            San Diego"
    },
    {
        "day": 6,
        "current_city": "San Diego",
        "transportation": "-",
        "breakfast": "Las Cuatro Milpas, San Diego",
        "attraction": "Balboa Park, San Diego;USS Midway Museum, San
            Diego;",
        "lunch": "Mitch's Seafood, San Diego",
        "dinner": "Puesto, San Diego",
        "accommodation": "Quaint And Cozy One Bedroom Unit With Patio,
            San Diego"
    },
    {
        "day": 7,
        "current_city": "from San Diego to Kona",
        "transportation": "Flight Number: HA, from San Diego to Kona,
            Departure Time: 11:17, Arrival Time: 14:19",
        "breakfast": "Kindred, San Diego",
        "attraction": "-",
        "lunch": "-",
        "dinner": "-",
        "accommodation": "-"
    }
```

### E.2 Demo

We present a comparative demonstration of the user-facing interface for monolithic agent baseline (Figure 11) vs. ATLAS (Figure 12), given the same real-world travel planning query: "Please create a travel plan for me where I'll be departing from Washington and heading to Myrtle Beach for a 3-day trip from November 13th to November 15th, 2025. Can you help me keep this journey within a budget of $1,400?". Demo video is available at https://www.youtube.com/watch?v=cy4KFd7GmRQ.

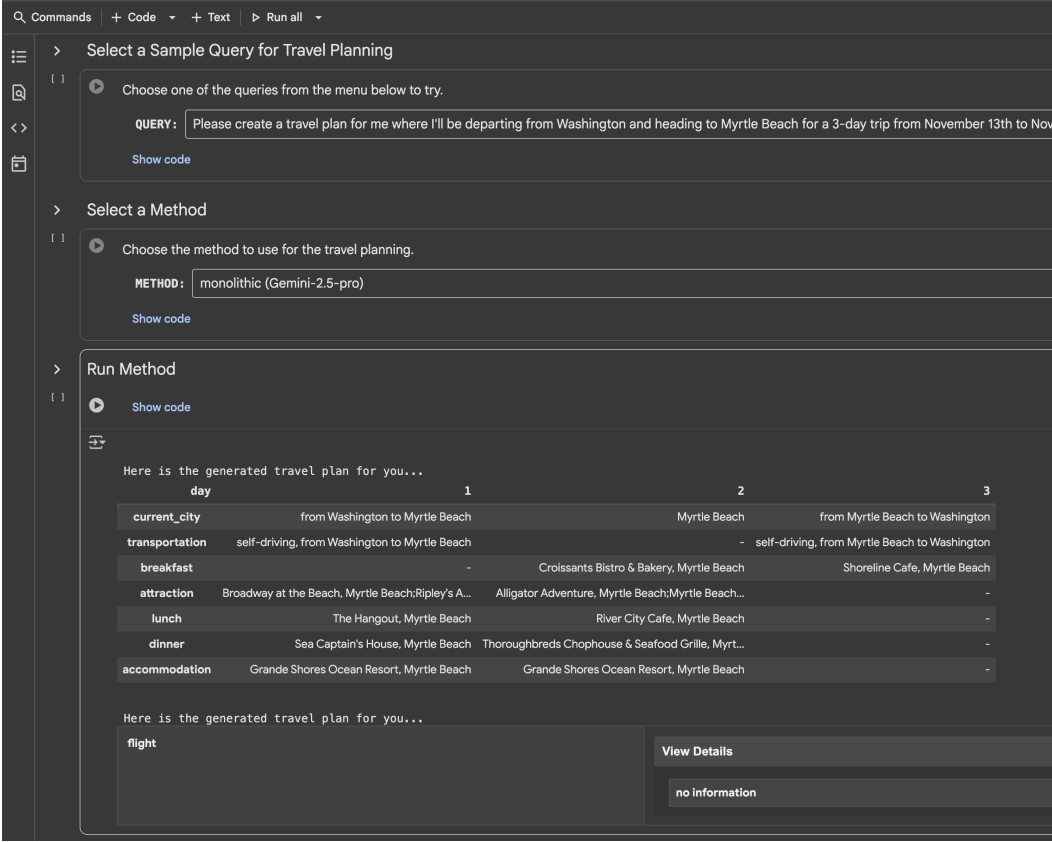

Figure 11: Screenshot of demo running monolithic agent for real-world travel planning. Displaying the generated plan and the summary of search results by the monolithic agent (Gemini-2.5-Pro).

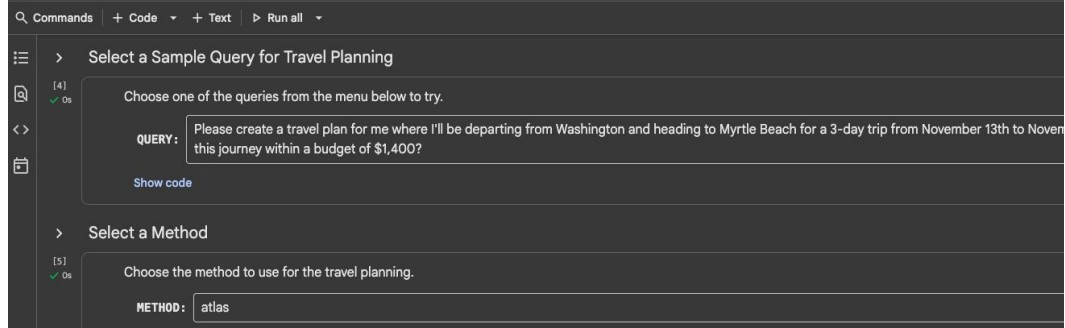

Running ATLAS and displaying generated plan

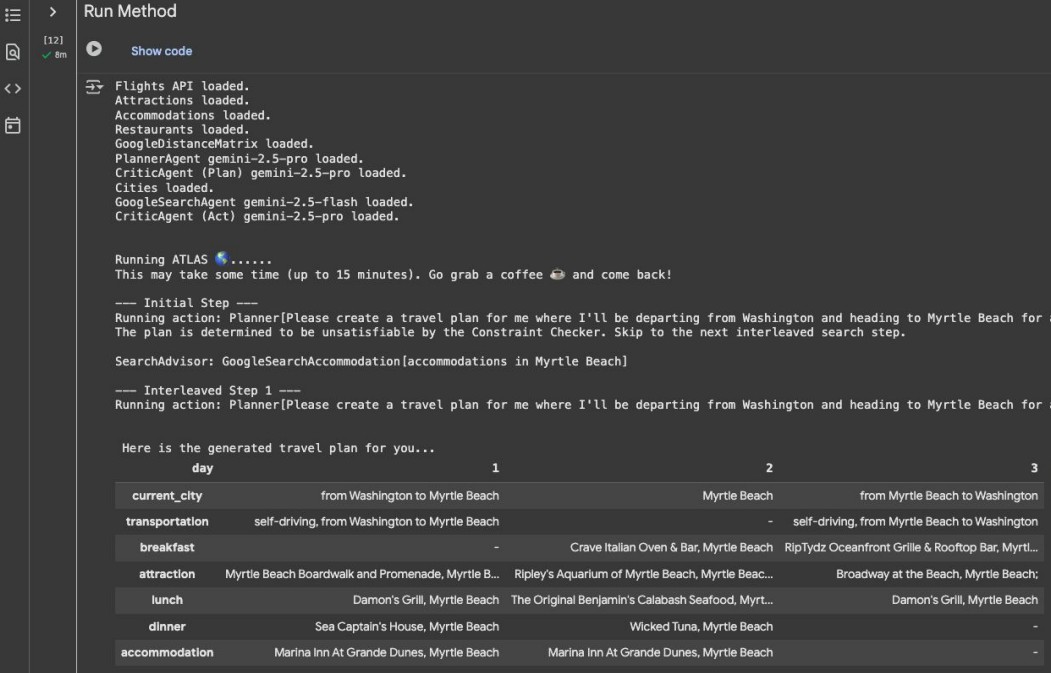

Displaying the summary of search results by ATLAS

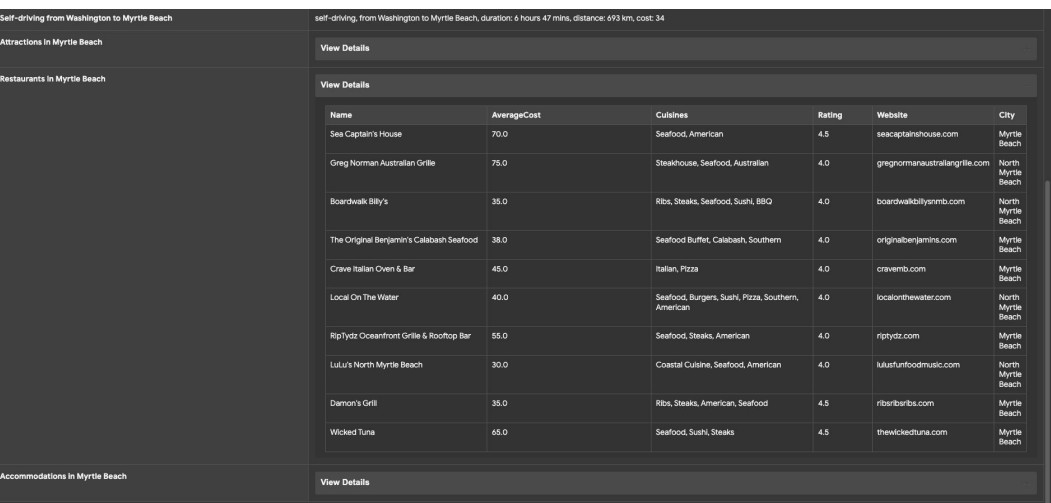

Figure 12: Screenshots of demo running ATLAS for real-world travel planning. Displaying the generated plan and the summary of search results by ATLAS.

# F PROMPTS

## F.1 TRAVELPLANNER

For the Search agent and the Planner agent, we basically follow the prompts as provided in Xie et al. (2024) for the search agent and the Planner agent [4]. We provide the full prompts for the agents in our framework: Constraint Manager, Checker [5], and Search Advisor. In their prompts, the list of tools and example plan are exactly as provided in the original prompts of the TravelPlanner benchmark.

---

**Prompt used for Constraint Manager**

```
You are an expert in logical reasoning whose task is to list out
    constraints that the user must adhere to when creating a travel
    plan.
Given a query AND context information,these constraints can be any
    relevant factors that may be explicitly identifiable from them.
    The query may explicitly specify some constraints, but you
    should also consider those that can be inferred from the
    context information. Do NOT include constraints that cannot be
    explicitly formulated from query and the context information;
    for example, do not consider the feasibility of the plan --
    i.e., whether the schedule is practical enough for the user to
    pull off.

Additional notes:
1. Use the city name exactly as provided; for instance,
    "Washington" refers to the one in Seattle, WA, not "Washington
    D.C". Do not infer a different city. Only use cities directly
    relevant to the query, and do not include nearby cities with
    airports or those in the general vicinity.
2. The trip must be a closed loop. Even though it is unclear from
    the user query whether it should return to the origin city,
    always assume that it should.

As your output, enumerate a list of constraints that the user must
    adhere to when creating a travel plan. Output must be in a
    structured format with numbered constraints. Keep it succinct
    and only list the constraints and do not add any additional
    statements. Prioritize the constraints that are not drawn from
    the query, but that are additionally specified from the context
    information.

query: [query inserted here]
context information: [context information inserted here]
constraints:
```

---

**Prompt used for Checker**

```
You are an expert in logical reasoning whose task is to act as a
    critic.
You are paired with a travel planner, and will be given a query and
    the plan generated by the travel planner.
```

---

[4]'ZEROSHOT_REACT_INSTRUCTION' and 'PLANNER_INSTRUCTION', respectively from here: https://github.com/OSU-NLP-Group/TravelPlanner/blob/main/agents/prompts.py

[5]The prompt for Checker includes sandbox-specific instructions (*e.g.,* "Note that transportation duration is a secondary concern...", "When it comes to accommodation, don't worry about maximum occupancy constraint.", "Implicit Allowance...", "Strict Filtering..."). This is because the sandbox data often has no feasible options when these constraints are not relaxed, and the evaluation metrics do not check for them. For a fair comparison, we provide these instructions to baseline methods as well. These instructions are removed for our LivePlanner setup, where search results are no longer limited by sandbox-limited feasibility.

You are given the context information, a collection of travel
    information that were referred to when the travel planner
    created the plan.
You are given constraints, which are a list of rules that the plan
    must comply with.
You are also given previous planning attempts and your feedback on
    them, which will provide you with holistic insights into the
    planning process so far.

Your goal is to evaluate the travel plan created by the planner,
    ONLY on the following aspects:
1. VERY IMPORTANT: every detail in the plan must be derived from
    the provided information.
2. VERY IMPORTANT: for each day, any necessary applicable details
    on transportation, restaurant, attraction, and accommodation
    should not be missing. Note that transportation duration is a
    secondary concern: While long transportation durations (e.g.,
    exceeding 20 hours) might occur, do not let them be the primary
    basis for rejecting a plan or skipping essential elements like
    accommodation.
3. Every part of the plan must adhere to the provided constraints,
    whenever applicable.
4. Each day permits the assignment of no more than one
    transportation method and one accommodation. All travelers must
    stay in the one accommodation together and not split up into
    multiple accommodations. When it comes to accommodation, don't
    worry about maximum occupancy constraint.
5. Overall travel sequence and all details in the plan should align
    with commonsense. However, when it comes to the feasibility of
    the plan, focus more on the completeness of the plan (e.g., are
    all necessary plan details included?) rather than meticulously
    scrutinizing the exact feasibility of transportation durations,
    for example. Small variations in travel time should not be a
    major point of criticism.

Make sure to only evaluate the travel plan based on the above
    aspects.
You must not evaluate on the format of the travel plan, as the
    planner is required to follow a specific format.
No transportation is needed when not moving between cities.
All price is for one person, and all accommodation price is per
    night.

When evaluating accommodation rules, please adhere to the following
    principles:
1. Implicit Allowance: Unless a rule explicitly states a
    prohibition, assume that the activity or feature is permitted.
    Do not infer restrictions based on the absence of explicit
    permission.
2. Strict Filtering for Prohibitions: When a user expresses a
    preference or requirement, strictly filter out any
    accommodations that explicitly state a prohibition against that
    preference. Avoid overthinking or applying overly broad
    interpretations to these prohibitions. Focus solely on direct
    contradictions.

Planning failures may stem from the planner's limitations in
    utilizing existing information, or from the incompleteness of
    the provided context information.
If it is the planner's fault, you should provide feedback on the
    specific reasons why the plan is invalid.
If it is the fault of the provided context information, you should
    identify the case as unsatisfiable and provide feedback on what

```
        further information was needed to make the plan valid. Or if it
        repeats the same failure type as previous planning attempts,
        you should also identify the case as unsatisfiable.

Output your critic results in a structured JSON format with two
    fields as in the example: (1) decision and (2) feedback.
(1) For decision, it must be valid, invalid, or unsatisfiable.
(2) When the decision is valid, do not provide any feedback. When
    it's invalid, enumerate the reasons for your decision. When the
    decision is unsatisfiable, it means that the plan cannot be
    successfully generated based on the provided context
    information mainly due to insufficient information. In this
    case, it is not you and the planner's fault, but rather has to
    be resolved by collecting more information, so you should
    provide feedback on what further information was needed to make
    the plan valid.
Keep your output concise and don't include suggestions for the
    improvement, focusing on the missing information in the travel
    plan or the constraints that are not satisfied.

***** Example 1 *****
Decision: invalid
Feedback: 1. The accommodation choice is missing for Day 1.
2. The plan includes a restaurant choice that is not in the
    provided context information.
3. The minimum nights for the accommodation 'Affordable Spacious
    Refurbished Room in Bushwick!, Charlotte' is 2, but it is only
    booked for 1 night on Day 1.
4. The plan violates the constraint that the mode of transportation
    must be self-driving for the entire trip, as a flight is chosen
    for Day 1.
5. The city sequence does not make sense.
***** Example 1 Ends *****

***** Example 2 *****
Decision: unsatisfiable
Feedback: The plan is invalid because it does not include any
    transportation for Day 1, which is necessary for the trip.
    However, this failure was because there is no transportation
    information for Day 1 in the provided context information,
    hence further information collection is required.
***** Example 2 Ends *****

query: [query insearted here]
context information: [context information inserted here]
constraints: [outputs from Constraint Manager inserted here]
previous planning attempts and your feedback: [previous planning
    attempts and corresponding feedback inserted here]
travel plan: [current plan to be validated inserted here]
critic:
```

**Prompt used for Search Advisor**

```
You are an expert in logical reasoning whose task is to act as a
    critic.
You are paired with an assistant who will take actions to collect
    information related to transportation, dining, attractions, or
    accommodation for planning a vacation trip. Each action by the
    assistant only calls one function once, each of which MUST be
    one of the following types:

[List of tools inserted here...]
```

```
*******

Let me also give you an example of a good travel plan, to inform
    you of the output format from the Planner. The symbol '-'
    indicates that information is unnecessary.

[Example plan insterted here...]

You will be given a query and the actions and observations taken so
    far. You are also given previous planning attempts made based
    on the observed information along with the feedback.
These planning attempts have failed and the accompanied feedback
    explains why they failed.
Now your task, as a critic, is to identify any gaps in the
    information collected so far and suggest additional actions to
    gather the necessary information. See if any of the previous
    planning failures are because that was the only possible plan
    outcome given the limited previous information, and the planner
    could have done better only if the assistant had collected more
    information.
Your output is the list of actions that should be taken for further
    comprehensive information collection and potentially to help
    address the previous planning failures. Remember, the actions
    should be one of the specified actions described above, and you
    should not suggest any actions that go beyond the scope of
    those actions or have already been taken.

IMPORTANT NOTES:
1. Your priority is to ensure that all comprehensive information on
    transportation, dining, attractions, and accommodation are
    collected and recorded in Notebook so that the Planner can use
    them.
2. Prevent redundant information gathering. Do not suggest calling
    actions that would collect information already present in the
    previously gathered data. If correct actions were taken but the
    observed information remains insufficient, do NOT repeatedly
    ask for similar actions until enough information is gathered.
3. If you believe all relevant information has been collected,
    suggest calling the Planner tool with the query. In this
    scenario, do not propose further actions, as previous planning
    failures might stem from the Planner's limitations in utilizing
    existing information, rather than a lack of it.
4. Use the city name exactly as provided; for instance,
    "Washington" refers to the one in Seattle, WA, not "Washington
    D.C". Or do not infer a different city; when calling actions,
    only use cities directly relevant to the query, and do not
    include nearby cities with airports or those in the general
    vicinity.
5. When gathering information on a specific topic from calling an
    action call, the presence of at least one relevant and
    satisfying piece of information is considered sufficient,
    meaning information collection for that topic is successful. It
    is not necessary for the majority of observed items to satisfy
    the query's specifications; the existence of a single suitable
    option is enough. The precise identification of that relevant
    piece within the observed information are the responsibility of
    the Planner, not the information extraction process.

Regarding accommodations, please adhere to the following principles:
1. Implicit Allowance: Unless a rule explicitly states a
    prohibition, assume that the activity or feature is permitted.
```

```
    Do not infer restrictions based on the absence of explicit
    permission.
2. Strict Filtering for Prohibitions: When a user expresses a
    preference or requirement, strictly filter out any
    accommodations that explicitly state a prohibition against that
    preference. Avoid overthinking or applying overly broad
    interpretations to these prohibitions. Focus solely on direct
    contradictions.

Remember, you want to efficiently gather all necessary information.
    You should not suggest actions that collect information that
    goes beyond the scope of the query or that is not relevant to
    the query.
Keep your output succinct.
Do not include any Action Number in your suggested actions: for
    example, if you suggest `GoogleDistanceMatrix[Twin Falls, Salt
    Lake City, self-driving]`, just output it directly, not in the
    form of `Action 1: GoogleDistanceMatrix[Twin Falls, Salt Lake
    City, self-driving]`.
```

F.2    LIVE TRAVEL PLANNING

For live travel planning, instead of the sandbox tools, we use the Google Search based tools to retrieve live information on flights, accommodations, restaurants, and attractions. For the Search agent, we use the same prompt as in TravelPlanner, but the list of tools is replaced with those in "Prompt listing the tools used for live travel planning". Search Advisor also uses the same prompt as in TravelPlanner, but the list of tools is replaced with those in "Prompt listing the tools used for live travel planning".

---

**Prompt listing the tools used for live travel planning**

```
(1) GoogleSearchFlight[Origin, Destination, Date]:
Description: A flight information retrieval tool that uses Google
    Search.
Parameters:
Origin: The city you'll be flying out from.
Destination: The city you aim to reach.
Date: The date of your travel in YYYY-MM-DD format.
Example: GoogleSearchFlight[New York, London, 2025-10-01] would
    fetch flights from New York to London on October 1, 2025.

(2) GoogleDistanceMatrix[Origin, Destination, Mode]:
Description: Estimate the distance, time and cost between two
    cities. DO NOT use this tool to find the transportation inside
    a city. Don't worry about the transportation inside a city as a
    part of your travel planning.
Parameters:
Origin: The departure city of your journey. It must be just a city
    name, not other names like airport name, without including
    state code, etc.
Destination: The destination city of your journey.  It must be just
    a city name, not other names like airport name, without
    including state code, etc.
Mode: The method of transportation. Choices include 'self-driving'
    and 'taxi'.
Example: GoogleDistanceMatrix[Paris, Lyon, taxi] or
    GoogleDistanceMatrix[Paris, Lyon, self-driving] would provide
    driving distance, time and cost between Paris and Lyon.

(3) GoogleSearchAccommodation[searchQuery]:
Description: Discover accommodations in your desired city using
    Google Search.
Parameters: searchQuery - the rephrased query that only includes
    necessary details about the accommodation search.
Example: GoogleSearchAccommodation[Find accommodations in Rome from
    2025-10-01 to 2025-10-05 for 2 guests. We require
    accommodations in the form of private rooms.] would present a
    list of accommodations in Rome from October 1 to October 5,
    2025, for 2 guests.

(4) GoogleSearchRestaurant[searchQuery]:
Description: Explore dining options in a city of your choice using
    Google Search.
Parameter: searchQuery - The rephrased query that only includes
    necessary details about the restaurant search.
Example: GoogleSearchRestaurant[Find restaurants in Tokyo. I want
    to try Korean and Japanese cuisines.] would show a curated list
    of restaurants in Tokyo.

(5) GoogleSearchAttraction[City]:
Description: Find attractions in a city of your choice using Google
    Search.
Parameter: City - The name of the city where you're seeking
    attractions.
```

---

```
Example: GoogleSearchAttraction[London] would return attractions in
    London.

(6) CitySearch[State]
Description: Find cities in a state of your choice.
Parameter: State - The name of the state where you're seeking
    cities.
Example: CitySearch[California] would return cities in California.

(7) NotebookWrite[Short Description]
Description: Writes a new data entry into the Notebook tool with a
    short description. This tool should be used immediately after
    FlightSearch, AccommodationSearch, AttractionSearch,
    RestaurantSearch or GoogleDistanceMatrix. Only the data stored
    in Notebook can be seen by Planner. So you should write all the
    information you need into Notebook.
Parameters: Short Description - A brief description or label for
    the stored data. You don't need to write all the information in
    the description. The data you've searched for will be
    automatically stored in the Notebook.
Example: NotebookWrite[Flights from Rome to Paris in 2022-02-01]
    would store the informatrion of flights from Rome to Paris in
    2022-02-01 in the Notebook.

(8) Planner[Query]
Description: A smart planning tool that crafts detailed plans based
    on user input and the information stored in Notebook.
Parameters:
Query: The query from user. Make sure that this is exactly the same
    query given from the user, not a paraphrased one.
Example: Planner[Give me a 3-day trip plan from Seattle to New
    York] would return a detailed 3-day trip plan.
You should use as many as possible steps to collect enough
    information to input to the Planner tool.

In the input arguments, use the city name exactly as provided; for
    instance, "Washington" refers to the one in Seattle, WA, not
    "Washington D.C". Or do not infer a different city. Only use
    cities directly relevant to the query, and do not include
    nearby cities with airports or those in the general vicinity.

Each action only calls one function once. Do not add any
    description in the action. Output only one action at a time.
Do NOT include your thought in the action output. Your action must
    be simply just calling one of the above eight actions.
Do not use the word 'Action' and the number in your output; for
    example, only output GoogleDistanceMatrix[El Paso, Phoenix,
    self-driving], not Action 10: GoogleDistanceMatrix[El Paso,
    Phoenix, self-driving].
```

**Prompt for live flight search**

```
site: www.google.com/travel/flights OR site:
    www.expedia.com/Flights.
Request: [request from the search agent inserted here]

Given the search results, extract all necessary flight information
    and output in the requested format.
The output should be in a structured format with the following
    columns: FlightNumber, Price, DepTime, ArrTime,
    ActualElapsedTime, FlightDate, OriginCityName, DestCityName.
```

```
Try not to miss any fields. For 'FlightNumber', it's okay to just
    use the airline code (like DL) if inevitable, without the full
    flight number (like DL5375).
No field should be left blank or None.

Prioritize the cheapest flights.
All price is for one person.

When there is no available flight option at all, you must return
    'no information'.

Here are the examples of the desired output. Value for each column
    should be clearly separated by a semicolon and a tab.
=== Example 1 begins ===
FlightNumber;    Price;    DepTime;    ArrTime;
    ActualElapsedTime;    FlightDate;    OriginCityName;
    DestCityName
F3502691;        240;       18:48;        20:51;      2 hours 3 minutes;
        2022-03-02;        Buffalo;      Atlanta
F3514187;        322;       06:51;        08:40;      1 hours 49
    minutes;       2022-03-02;        Buffalo;       Atlanta
F3555201;        265;       12:44;        14:33;      1 hours 49
    minutes;       2022-03-02;        Buffalo;       Atlanta
=== Example 1 ends ===

=== Example 2 begins ===
no information
=== Example 2 ends ===

DO NOT include anything else but only the collected information
    exactly structured as requested above.
Formatted output:
```

**Prompt for live accommodation search**

```
site: www.expedia.com/Hotels OR site: www.airbnb.com OR site:
    www.booking.com.
Request: [request from the search agent inserted here]

Given the search results, extract all necessary accommodation
    information and output in the requested format.
The output should be in a structured format with the following
    columns: name, price, maximum_occupancy, rating, city.
Try not to miss any fields, but if inevitable, it's okay to leave
    the 'maximum_occupancy' and 'rating' fields to be None.
Prioritize the cheapest accommodations.

When there is no available accommodation option at all, you must
    return 'no information'.
All price is for one person per night.

Here are the examples of the desired output. Each field should be
    separated by a semicolon and a tab.
=== Example 1 begins ===
name;       price;       maximum_occupancy;       rating;       city
Hilton Hotel;       212.0;       2;       3.0;       Tucson
Marriott Marquis;       357.0;       2;       5.0;       Tucson
Green Oasis;       118.0;       2;       3.0;       Tucson
Beacon Grand;       58.0;       2;       3.0;       Tucson
Sunny Cobble Hill;       107.0;       3;       2.0;       Tucson
Hotel Zetta;       231.0;       2;       5.0;       Tucson
=== Example 1 ends ===
```

```
=== Example 2 begins ===
no information
=== Example 2 ends ===

DO NOT include anything else but only the collected information
    exactly structured as requested above.
Formatted output:
```

**Prompt for live attraction search**

```
site: www.tripadvisor.com/Attractions
Request: [request from the search agent inserted here]

Given the search results, extract all necessary attraction
    information and output in the requested format.
The output should be in a structured format with the following
    columns: name, address, phone, website, city. No field should
    be left blank or None.
Please try to return at least 6 restaurants.
Prioritize the most popular attractions.

When there is no available attraction option at all, you must
    return 'no information'.

Here are the examples of the desired output. Each field should be
    separated by a semicolon and a tab.
=== Example 1 begins ===
Name;      Address;      Phone;      Website;      City\n
The Dallas World Aquarium;    1801 N Griffin St, Dallas, TX
    75202, USA;      (214) 720-2224;      https://www.dwazoo.com/;
         Dallas\n
The Sixth Floor Museum at Dealey Plaza;      411 Elm St, Dallas, TX
    75202, USA;      (214) 747-6660;      https://www.jfk.org/;
     Dallas\n
Reunion Tower;      300 Reunion Blvd E, Dallas, TX 75207, USA;
    (214) 296-9950;      http://www.reuniontower.com/;      Dallas\n
Dallas Museum of Art;      1717 N Harwood St, Dallas, TX 75201,
    USA;      (214) 922-1200;      https://www.dma.org/;      Dallas
=== Example 1 ends ===

=== Example 2 begins ===
no information
=== Example 2 ends ===

DO NOT include anything else but only the collected information
    exactly structured as requested above.
Formatted output:
```

**Prompt for live restaurant search**

```
site: www.tripadvisor.com/Restaurants
Request: [request from the search agent inserted here]

Given the search results, extract all necessary restaurant
    information and output in the requested format.
The output should be in a structured format with the following
    columns: Name, AverageCost, Cuisines, Rating, and City.
For the average cost, if there is no direct price information,
    depending on the price descriptions or dollar signs, and the
```

```
      living cost of the city, you MUST determine the average cost as
      a specific number.
Try not to miss any fields, but if inevitable, it's okay to leave
      the `Rating` field to be None.
Prioritize the most popular restaurants or highly rated restaurants.
Please try to return at least 10 restaurants. If there is a
      specific request for cuisines, try to return at least 4
      restaurants of each cuisine.

When there is no available restaurant option at all, you must
      return `no information`.
All price is for one person.

Here are the examples of the desired output. Each field should be
      separated by a semicolon and a tab.
=== Example 1 begins ===
Name;       AverageCost;     Cuisines;      Rating;      City
Coconuts Fish Cafe;      97.0;       Mediterranean;      4.5;
      Dallas
1918 Bistro & Grill;      87.0;       BBQ Seafood;      4.4;
      Dallas
Yanki Sizzlers;      56.0;       Cafe French;      4.1;      Dallas
Aravali Owls;      29.0;      Italian;      4.7;      Dallas
=== Example 1 ends ===

=== Example 2 begins ===
no information
=== Example 2 ends ===

DO NOT include anything else but only the collected information
      exactly structured as requested above.
Formatted output:
```

## 6 Large Language Model Usage for Writing

In this paper, we used LLMs, specifically Gemini and ChatGPT, strictly only as general-purpose writing tools. We provided draft text asking the models to correct any grammatical errors, refine the structure, or reduce the redundancy. All edited text was then manually verified and edited as needed. The LLMs were not used for generating any new content or references.

