# OpenReview forum: "ATLAS: Constraints-Aware Multi-Agent Collaboration for Real-World Travel Planning"
_ICLR.cc/2026/Conference — ICLR 2026 Poster_

### Official Review · Reviewer_LbP3 · 2025-10-26

**Soundness:** 4
**Presentation:** 4
**Contribution:** 3
**Rating:** 8
**Confidence:** 4

**Summary:**

This paper presents ATLAS, a multi-agent framework designed to tackle travel planning, which requires reasoning under complex, evolving, and often implicit constraints. ATLAS formalizes the task as a dynamic constraint satisfaction problem and introduces two cooperative agents, including a constraint manager for explicit and implicit constraint construction and a planner-checker pair for iterative plan generation and validation. Through such a multi-agent system design, ATLAS continuously refines plans, resolving information gaps that  single-agent systems often fail to address. Experiments on the TravelPlanner and Flex-TravelPlanner benchmarks show that ATLAS consistently achieves state-of-the-art performance, surpassing strong baselines such as ReAct, Reflexion, and PMC.

**Strengths:**

1. The paper is grounded in a clear and compelling motivation The authors effectively articulate why constraint awareness and multi-agent collaboration are essential for complex planning scenarios, making the research direction both timely and impactful.
2. The experimental evaluation is extensive and rigorous. The authors conduct detailed comparisons with several strong baselines (e.g., ReAct, Reflexion, EvoAgent, and PMC) and perform ablation studies that carefully examine the contribution of each module within ATLAS, such as the constraint manager and checker. The results are analyzed with clear quantitative evidence, supported by thoughtful discussions and visualizations.
3. The test environments go beyond the standard TravelPlanner sandbox benchmark. The introduction of Flex-TravelPlanner for multi-turn interaction and the integration of live web search in real-world scenarios substantially enhance the ecological validity of the evaluation. These settings convincingly demonstrate ATLAS’s robustness, adaptability, and practical potential for real-world deployment beyond controlled benchmark conditions

**Weaknesses:**

The only concern is that, after reviewing the prompts provided in the appendix, the framework appears to require a considerable amount of manual effort to design suitable prompts for different agents in each specific scenario. While this is acceptable given that the paper focuses solely on travel-planning tasks, such dependence on handcrafted prompts may limit ATLAS’s generalization and scalability to other domains or broader real-world applications.

**Questions:**

1. In Table 10 (Appendix D.3.2), it is somewhat surprising that Gemini-2.5-Pro achieves better performance on the hard set than on the medium set. I am curious about this observation, as it appears that only ATLAS exhibits such an uncommon pattern.

---

> ### Author Response · Authors · 2025-11-22
> **Response to Reviewer LbP3**
>
> Dear Reviewer LbP3,
>
> We sincerely thank you for your careful reading of our paper and your encouraging feedback. We are encouraged by your strong endorsement regarding our problem formulation and the extensiveness of our evaluations.\
> Please find our responses to your comments below.
>
> &nbsp;
> ## Generalization and Manual Prompt Effort
> We acknowledge your point regarding the manual effort involved in designing prompts. While we carefully designed the prompts to be as general as possible within the travel domain (avoiding metric-specific hacking), we agree that adapting the framework to entirely new domains currently requires rewriting agent/tool specifications. To mitigate this, we envision that ATLAS can be integrated with recent advancements in automated prompt optimization. Frameworks like DSPy [1] or TextGrad [2] can automate the synthesis and tuning of specific instructions for each agent, significantly reducing human effort. This automation will be beneficial when extending our ATLAS framework to new, information-rich, and constraint-heavy domains such as agent guardrails with user-defined safety policies or personalization agents (see line 485).
>
> [1] Khattab et al., "DSPy: Compiling Declarative Language Model Calls into Self-Improving Pipelines", ICLR 2024. https://openreview.net/pdf?id=sY5N0zY5Od \
> [2] Yuksekgonul et al., "Optimizing generative AI by backpropagating language model feedback", Nature 2025. https://arxiv.org/pdf/2406.07496
>
> &nbsp;
> ##  Performance on "Medium" vs. "Hard" Subsets
> We appreciate your keen observation! You are correct that ATLAS performs better on the "hard" set than the "medium" set. We investigated this phenomenon and found that it is consistent with the inherent characteristics of the TravelPlanner benchmark, rather than an anomaly specific to ATLAS. For example, this trend is observable in Table 4 of the original TravelPlanner paper [3], where pass rates for "medium" are often lower than "hard" for both two-stage and sole-planning setups. Similarly, Table 3 of the PMC baseline [4] reports a final pass rate for "medium" that is ~8% lower than "hard". In our Table 10, the baseline ReAct also exhibits this pattern (Medium: 11.67% vs. Hard: 21.67% final pass rate).
>
> One possible hypothesis to explain this counter-intuitive observation is the trade-off between constraint complexity and domain set size (search space) in the constraint satisfaction problem.
> * Easy to medium/hard: The increase in constraint complexity (e.g., adding constraints beyond just budget) is the dominant factor, making the task strictly harder.
> * Medium to hard: While "hard" tasks impose more constraints (adding transportation preferences on top of "medium" constraints), these specific constraints act as strong filters that prune the domain of applicable search results significantly. This reduces the number of "distractor" options for the planning agent, making the correct assignment of values for variables clearer.
>
> If you see Table 4 of the TravelPlanner paper [3], as transportation preference constraints are added from medium to hard, there is a substantial increase in pass rates for transportation-relevant criteria (e.g., “Non-conflicting Transportation” rises from 73.3% to 83.1% in two-stage, and 56.5% to 87.5% in sole-planning settings). \
> We have extended Appendix D.6 (ablations on travel days and levels) with this discussion.
>
> [3] Xie et al., “TravelPlanner: A Benchmark for Real-World Planning with Language Agents”, ICML 2024. https://arxiv.org/pdf/2402.01622 \
> [4] Zhang et al., “Planning with Multi-Constraints via Collaborative Language Agents”, COLING 2025. https://aclanthology.org/2025.coling-main.672.pdf

---

### Official Review · Reviewer_Qh1s · 2025-10-31

**Soundness:** 2
**Presentation:** 2
**Contribution:** 2
**Rating:** 4
**Confidence:** 4

**Summary:**

This paper proposes ATLAS, a multi-agent framework formalized as a Constraint Satisfaction Problem, to address LLMs’ limitations in handling complex constraints for real-world travel planning. The framework comprises five agents (Search Agent, Constraint Manager, Planner, Checker, Search Advisor) targeting three challenges: constraint construction (explicit/implicit constraint extraction), constraint-aware planning (valid scheme generation), and information gap resolution (adaptive supplementary search).

**Strengths:**

1. ATLAS offers a structured approach to constraint discovery and information search, providing valuable heuristic insights.

2. Evaluations are conducted in real-world settings, incorporating live search (via Google Search instead of sandboxed tools) and reporting hallucination rates, demonstrating practical applicability.

3. The method achieves state-of-the-art performance on TravelPlanner, highlighting its effectiveness.

4. The algorithm is clearly described, the writing is accessible.

**Weaknesses:**

1.The framework, while well-engineered, largely repackages existing multi-agent and verification ideas. The plan–check–revise–search pipeline resembles prior systems like CRITIC, PMC, and LLM-Modulo Planning.

2.Although the task is formalized as a CSP, the link between classical CSP algorithms and the actual implementation remains superficial.

3.Evaluation is predominantly limited to TravelPlanner, raising concerns about overfitting and limited generalizability.

4.The multi-agent design lacks clear justification. While multi-agent systems often boost performance in travel planning, the specific agent roles and structures here appear arbitrary. Comparisons with more baselines (e.g., LLM-Modulo, verifier-aided approaches) and evaluations across diverse models (including open-source ones) and benchmarks are needed to demonstrate generality.

**Questions:**

Q1: What is the relationship between the multi-agent design and the CSP formalization?

Q2: Could similar improvements be attained by enhancing single-agent strategies (e.g., ReAct with verifier loops)?

Q3: Why does the algorithm perform well in multi-turn settings where others fail?

Q4: How would ablating specific agents impact performance?

---

> ### Author Response · Authors · 2025-11-22
> **Response to Reviewer Qh1s (1/4)**
>
> Dear Reviewer Qh1s,
>
> We sincerely appreciate your thoughtful review and constructive feedback. We address your concerns and questions below.
>
> &nbsp;
>
> ## Novelty (W1)
>
> We acknowledge that the technical instantiation of our framework is inspired by successful foundational concepts in the literature (lines 51-82).
> However, ATLAS goes beyond simple "repackaging" by addressing the specific limitations of existing methods in open-world, constraint-heavy environments:
>
> * (1) Beyond fact verification (vs. CRITIC): CRITIC primarily focuses on verifying external facts (e.g., ``is the population correct?’’). ATLAS’s Checker verifies logical consistency against a complex, dynamically evolving set of constraints (both explicit and implicit).
>
> * (2) Dynamic constraint construction (vs. LLM-Modulo): Approaches involving external formal solvers typically require a pre-defined solver schema or prompt that knows the constraints a priori.  For example, in Hao et al. [1], when translating user queries into code for formal solvers, it uses prompts with hardcoded constraints and hints on evaluation criteria: *“Assert first transportation happens at first day... get accommodation minimum night info and assert it to be less than the day's stay...”* (Appendix G.1.1 in [1]). In contrast, ATLAS does not start with a predefined list of constraints to look for. Relying on prior benchmark-specific knowledge (as in [1]) constitutes a form of evaluation metric leakage that simplifies the task from constraint discovery to merely variable filling, and makes the implementation overfit to a certain benchmark.
>
> * (3) Systemic decomposition (vs. PMC): Standard multi-agent systems (like PMC) typically rely on LLM-based task decomposition and still fall short in performance (achieving only 23.33% vs. ATLAS's 44.44% in Table 1). Our contribution is not just the combination of modules, but the systematic decomposition to overcome the current limited capabilities of monolithic LLMs.

---

> ### Author Response · Authors · 2025-11-22
> **Response to Reviewer Qh1s (2/4)**
>
> ## Relationship between our multi-agent design and CSP formalization (W2, Q1)
> > Although the task is formalized as a CSP, the link between classical CSP algorithms and the actual implementation remains superficial.
>
> We kindly direct the reviewer to `Appendix B` for a detailed discussion on the connection to CSP literature. On top of that, we clarify that our multi-agent architecture is not arbitrary; it is an instantiation of the CSP components defined in Section 2 ($P = \langle X, D, C \rangle$). The CSP formalization guided the identification of three fundamental challenges (i.e., (2) Constraint Construction, (3) Planning under Constraints, and (4) Resolving Information Gaps) and directly informed the design of a multi-agent orchestration system explicitly built to tackle them:
>
> * (1) Constructing Domain ($D$) → Search Agent:\
> In a classical CSP, the domain of values for variables is given. In real-world planning, the domain (e.g., available flights, hotels) is unknown and vast. The **Search Agent** is designed solely to populate $D$ by retrieving external information, transforming the open world into a structured set of candidate values for the Planner.
>
> * (2) Constructing Constraints ($C$) → Constraint Manager:\
> The foremost challenge in open-world CSP is identifying the set of constraints $C$ without any prior knowledge of benchmark-specific evaluation criteria (lines 52, 317, 727-728). Unlike LLM-modulo approaches that hardcode these criteria (as noted above regarding [1]), our **Constraint Manager** is the functional equivalent of a "Constraint Acquisition" module in CSP literature, responsible for explicitizing $C$ from the user query and search results.
>
> * (3) Satisfaction and Verification ($X \rightarrow D, V$) → Planner and Checker:\
> In CSP, a solver finds an assignment $\sigma: X \rightarrow D$ and verifies it ($V$).
> Similarly, **Planner** proposes an assignment $\sigma$ and **Checker** verifies if $\sigma \nvDash C$. We deliberately choose Natural Language (NL) as the logic layer for the Checker rather than a symbolic solver (e.g., Z3) because NL captures the nuance of implicit commonsense rules better than rigid symbolic schemas in a zero-shot setting. We also note the potential to augment the Checker with CSP solvers for formal rigor (Section 5), where the primary challenge remains the reliable translation of open-world constraints into formal representations [2], which we leave as a future extension.
>
> * (4) Resolving UNSAT via Domain Expansion → Search Advisor and Interleaved Search:\
> In our real-world travel planning setup involving search, an UNSAT verdict often stems from an "Information Gap" (incomplete $D$ in (1)). This directly informs the design of our **Interleaved Search** and the **Search Advisor** agent. Its specific role is to diagnose *why* the CSP is unsatisfiable and to recommend actions to expand $D$ thereby effectively closing the loop between the CSP solving and the environment.
>
> [1] Large language models can solve real-world planning rigorously with formal verification tools. Hao et al, 2025. \
> [2] Planning anything with rigor: General-purpose zero-shot planning with llm-based formalized programming. Hao et al, 2025
>
>
> &nbsp;
> ## Generalizability and Overfitting (W3)
> > Evaluation is predominantly limited to TravelPlanner, raising concerns about overfitting and limited generalizability.
>
> We address this concern by highlighting three key aspects of our evaluation strategy:
> * Domain-agnostic task decomposition: As detailed in our previous responses, ATLAS is derived from a general CSP formulation. The resulting orchestration mechanism is suitable for any information-rich planning task with complex constraints that can be abstracted into the same CSP formalism (lines 484-485).
>
> * Rigorous prompt design: To prevent overfitting, we strictly avoided hardcoding benchmark-specific evaluation metrics into our prompts. Our ablation (Appendix D.2.2) confirms that ATLAS achieves SOTA performance (44.44%) *without metric hacking* or specific hints. Notably, performance improves further to 59.44% when such benchmark-specific hints are added (following the evaluation setup in the PMC baseline), but our main results rely solely on robust, generalized reasoning.
>
> * While travel planning is the core scope of this work, we extend our evaluation significantly beyond the sandbox. By testing on multi-turn scenarios (Section 4.2) and the live open web (Section 4.3), we demonstrate that ATLAS generalizes effectively to the complex, dynamic nature of practical real-world settings.

---

> ### Author Response · Authors · 2025-11-22
> **Response to Reviewer Qh1s (3/4)**
>
> ## Ablating on individual contribution of each agent (W4, Q4)
>
> > While multi-agent systems often boost performance in travel planning, the specific agent roles and structures here appear arbitrary.
>
> To demonstrate that our agent roles are not arbitrary but essential for handling specific failures, we performed a granular ablation on the TravelPlanner validation set in addition to our existing ablation study in `Figure 3` and `Appendix  D.2.1`.
>
> **Table A: Detailed Component Ablation**\
> We isolate the search subtask by comparing our agent against a "Gold Search" oracle (100% recall, provided by the benchmark as “sole-planning” mode). We isolate Planning/Constraint subtasks by toggling the Checker and Constraint Manager modules.
> | Index | Search Source | (+Interleaved) | (+Check) | (+Constraint Manager) | Final Pass Rate|
> | :--- | :---: | :---: | :---: | :---: | :--- |
> | A-1 | Gold Search | - | x | x | 31.67 |
> | A-2 | Gold Search | - | o | x | 32.78 |
> | A-3 | Gold Search | - | o | o | 44.44 |
> | B-1 | LLM Search | x | x | x | 17.78 |
> | B-2 | LLM Search | x | o | x | 25.00 |
> | B-3 | LLM Search | x | o | o | 32.22 |
> | C-2 | LLM Search | o | o | x | 30.00 |
> | C-3 | LLM Search | o | o | o | 44.44 |
>
>
> &nbsp;
> ### Component Analysis
>
> (1) Quantifying Subtask Failures
> * Planning failure: Even with perfect information (Gold Search, A-1), the base planner only achieves a 31.67% pass rate. This indicates that pure planning failures (without guidance) are the dominant error source, contributing the largest gap from 100%.
> * Search failure: Comparing the best non-interleaved model (B-3) with its Gold counterpart (A-3) reveals that initial search failures contribute to a ~12% drop in performance (44.4% to 32.2%). This quantifies the specific impact of the search subtask failing to retrieve necessary context.
>
> (2) Successful Mitigation of Failures
> * Constraint Management and Checking: Introducing these subtasks consistently improves the planner's success rate. In the Gold setting, they recover 12.77% of failures (A-1 → A-3), proving their effectiveness in handling the "constraint awareness" subtask.
>
> * Row C-3 introduces the Interleaved Search (guided by the Search Advisor). This module recovers the performance loss completely, raising the pass rate from 32.22% back to 44.44%—matching the Gold Search performance.
>
>
> &nbsp;
> ### Additional results with different base model
> > Comparisons with more baselines (e.g., LLM-Modulo, verifier-aided approaches) and evaluations across diverse models (including open-source ones) and benchmarks are needed to demonstrate generality.
>
> To prove our architecture provides structural benefits across various choices of base models, we ran additional experiments with OpenAI’s GPT-5 (released after Gemini-2.5-pro and Claude-Sonnet-4, the baselines already considered in our submission). Experiments were run on a randomized subset of the validation set (10 data points: easy, medium, and hard 3-day itineraries).
>
> The results are as follows (all in %):
> | Methods | Delivery | Commonsense (Micro / Macro) | Hard (Micro / Macro) | Final Pass |
> | :--- | :--- | :--- | :--- | :--- |
> | Planning without search (sole-planning) | 100.00 | 95.83 / 70.00 | 71.43 / 46.67 | 33.33 |
> | Planning with search (ReAct) | 100.00 | 89.58 / 43.33 | 50.00 / 46.67 | 26.67 |
> | Planning with search (ATLAS) | 100.00 | 93.33 / 56.67 | 67.29 / 60.00 | 49.00 |
>
> * The Search Bottleneck Persists: Similar to the observation with Gemini-2.5-pro, we observe a significant drop from 33.33% (sole-planning) to 26.67% (ReAct) when forced to search for information. This confirms the "Information Gap" is a fundamental challenge, not a model capability issue.
> * ATLAS Improvement: ATLAS boosts GPT-5’s performance to 49.00%, significantly outperforming both the ReAct baseline and the Sole-planning setup. This demonstrates that our orchestration framework is complementary to the reasoning capabilities of SOTA models.
>
> We acknowledge the value of extending our evaluation to open-source models. Due to the limited timeframe and deployment resources, we prioritize other SOTA reasoning models in this response. We appreciate the suggestion and view it as a meaningful future extension. \
>
> Moreover, in addition to TravelPlanner, Flex-TravelPlanner, and our Live Search setup, we are actively evaluating on the ChinaTravel benchmark [3] to further validate ATLAS's effectiveness on diverse constraint types. We will follow up with results shortly.
>
> [3] Shao et al.,  ChinaTravel: An Open-Ended Benchmark for Language Agents in Chinese Travel Planning, 2025.

---

> ### Author Response · Authors · 2025-11-22
> **Response to Reviewer Qh1s (4/4)**
>
> ## Comparison with single-agent + verifier loop (Q2)
> > Could similar improvements be attained by enhancing single-agent strategies (e.g., ReAct with verifier loops)?
>
> We examined this directly using the ablation data (from Table A in the above response [3/4]).
> * ReAct (Base): 17.78% (Row B-1)
> * ReAct + Checker loop: 25.00% (Row B-2) – Improves, but limited.
> * ReAct + Checker + Constraint Manager: 32.22% (Row B-3) – Better, but information gap still remains as a major bottleneck.
> * ATLAS (Full with Interleaved Search): 44.44% (Row C-3) – best performance.
> Simply adding a verifier loop to ReAct provides some gain, but it plateaus because the single agent struggles to maintain the global state of domain and constraints while also executing planning steps. ATLAS’s specialized decomposition ensures that constraints are explicitly managed and information gaps are actively resolved, yielding significantly higher performance.
>
>
> &nbsp;
>
> ## Reason for ATLAS’s good performance in multi-turn settings (Q3)
> > Why does the algorithm perform well in multi-turn settings where others fail?
>
> The key factor is ATLAS's explicit maintenance of a persistent state for Domains ($D$) and Constraints ($C$), rather than relying solely on unstructured conversation history. While standard agents like ReAct suffer from *context drift* (i.e., forgetting earlier rules or hallucinating details as the context grows), ATLAS caches search context $D$ and explicitly keeps track of evolving constraint sets and planning attempts at each turn.

---

> ### Comment · Reviewer_Qh1s · 2025-11-28
>
> I think the author has well resolved my confusion, and I will raise the score.
> Opus, I noticed that the edit button has disappeared. If it is restored later, I will raise score. If not, I support accepting this paper. It is an excellent paper in terms of both the design of the overall multi-agent algorithm and the final performance improved presented.

---

> > ### Author Response · Authors · 2025-11-29
> >
> > We are delighted to hear that our response has resolved your concerns. We sincerely thank you for acknowledging the value of our multi-agent orchestration design and the significant performance gains it demonstrates.
> > While we note the current system restrictions regarding score updates, we are deeply grateful for your endorsement and support for the paper's acceptance.

---

### Official Review · Reviewer_i4Up · 2025-11-01

**Soundness:** 3
**Presentation:** 4
**Contribution:** 3
**Rating:** 6
**Confidence:** 4

**Summary:**

- The paper proposes **ATLAS**, an agentic framework for real‐world travel planning that handles several constraints.
- The authors conduct **experiments** under multiple settings, including single-agent, multi-agent, and multi-agent with live search, showing robustness across settings.
- Their experiments demonstrate that ATLAS effectively reduces hallucinations and improves planning reliability compared to existing baselines.

**Strengths:**

Overall, the paper is well written and covers a comprehensive range of experiments. Here are some breakdowns:

- **Problem design / framework:** Problem settings are well estabilshed, and they are grounded in classic planning research.
- **Writing:** Well organized and easy to follow.
- **Experiments:** Comprehensive experiments, divided into single, multi, and multi+live settings.
- **Results:** Effectively addresses hallucination, which is a major issue, and shows remarkable improvements compared to baselines.
- **Analysis:** Provides a comprehensive failure analysis with fine-grained breakdowns.

**Weaknesses:**

### Weaknesses that may affect overall assessment

**W1. Lack of analysis**

> Related part: Section 4, Figure 3
>
- Although ATLAS demonstrates strong performance,  it’s crucial to show *why* a method works or fails — that would represent an important contribution.
- While Figure 3 provides a weak form of analysis, ATLAS follows an agentic workflow with multiple subtasks (e.g., search, plan). It would be valuable to show the success rate of each subtask and how failures in specific stages contribute to overall failure. There could even be interesting cases where later stages compensate for earlier failures.
- Different models may also show varying strengths across subtasks. Understanding *which model excels at which component* could guide future research into multi-agent workflows for more powerful travel-planning agents.

**W2. Lack of baseline comparison**

> Related part: W1, Appendix A
>
- Although the authors mention the paper [1] and note that their problem scope differs, I still think a comparison is needed. (Or with other sand-box based framework)
- I understand [1] *operates in a sandbox-based setup,* but it remains a strong tool achieving top performance. **The authors could, for example, disable ATLAS’s search component or provide the same searched data to [1] and compare outputs.

---

### Weaknesses that likely affect but not majorly

**W3. Lack of efficiency analysis**

> Related section: 3.4
>
- The authors claim ATLAS is efficient due to information reuse. A cost analysis (e.g., API cost comparison with single-model baseline) would strengthen this claim.

---

### Weaknesses unlikely to affect assessment but useful suggestions

**W4. Consideration of “preferences”**

- Recently, several studies [2,3] have explored preference constraints.
- If the authors show that ATLAS performs well under such conditions, the paper would be even stronger.
---
Reference:

    [1] Large language models can solve real-world planning rigorously with formal verification tools. Hao et al, 2025.

    [2] COMPASS: A Multi-Turn Benchmark for Tool-Mediated Planning & Preference Optimization, Qin et al, 2025

    [3] Flex-TravelPlanner: A Benchmark for Flexible Planning with Language Agents, Oh et al, 2025

**Questions:**

**Q1 Regarding INTERLEAVED SEARCH**

- INTERLEAVED SEARCH seems to play an important role, in what proportion of test cases was it actually triggered?

---

> ### Author Response · Authors · 2025-11-22
> **Response to Reviewer i4Up (1/3)**
>
> Dear Reviewer i4Up,
>
> We sincerely thank you for recognizing our problem formulation and the comprehensiveness of our evaluations and analysis. We appreciate your constructive feedback on deepening the analysis even further. Please find our responses to your comments below.
>
> &nbsp;
> ## Component Analysis (W1)
>
> We agree with the reviewer that analyzing per-module success and error propagation is crucial. To address the reviewer’s request—quantifying subtask success and showing how later stages compensate for earlier failures—we conducted a detailed ablation on the TravelPlanner validation set (with Gemini-2.5-Pro as the base LLM).
>
> We isolate the search subtask by comparing our agent against a "Gold Search" oracle (100% recall, provided by the benchmark as “sole-planning” mode). We isolate Planning/Constraint subtasks by toggling the Checker and Constraint Manager modules.
>
> | Index | Search Source | (+Interleaved) | (+Check) | (+Constraint Manager) | Final Pass Rate|
> | :--- | :---: | :---: | :---: | :---: | :--- |
> | A-1 | Gold Search | - | x | x | 31.67 |
> | A-2 | Gold Search | - | o | x | 32.78 |
> | A-3 | Gold Search | - | o | o | 44.44 |
> | B-1 | LLM Search | x | x | x | 17.78 |
> | B-2 | LLM Search | x | o | x | 25.00 |
> | B-3 | LLM Search | x | o | o | 32.22 |
> | C-2 | LLM Search | o | o | x | 30.00 |
> | C-3 | LLM Search | o | o | o | 44.44 |
>
> &nbsp;
> ### Quantifying Subtask Failures
> * Planning failure: Even with perfect information (Gold Search, A-1), the base planner only achieves a 31.67% pass rate. This indicates that pure planning failures (without guidance) are the dominant error source, contributing the largest gap from 100%.
> * Search failure: Comparing the best non-interleaved model (B-3) with its Gold counterpart (A-3) reveals that initial search failures contribute to a ~12% drop in performance (44.4% to 32.2%). This quantifies the specific impact of the search subtask failing to retrieve necessary context.
>
> ### Success of Mitigation Subtasks
> * Constraint Management and Checking: Introducing these subtasks consistently improves the planner's success rate. In the Gold Search setting, they recover 12.77% of failures (A-1 → A-3), proving their effectiveness in handling the "constraint awareness" subtask.
>
> ### How later stages compensate for earlier failures
> In row C-3, we enable the Interleaved Search (a later stage), and this module recovers the performance loss significantly (raising 32.22% (B-3) back to 44.44% (C-3)).
> We also refer the reviewer to our answer to Q1 for more further detailed analysis on how often interleaved search is triggered.
>
> &nbsp;
> We have included the above detailed analysis in Appendix D.2 of the revised manuscript. We also agree that the reviewer’s comment on further analyzing the impact of base model choice on the analysis will be a valuable guide on future development of this multi-agent system. We will include that in the final manuscript.

---

> ### Author Response · Authors · 2025-11-22
> **Response to Reviewer i4Up (2/3)**
>
> ## Lack of baseline comparison (W2)
> We emphasize that our evaluation already covers a comprehensive range of baselines adapted for our ``planning with search’’ setting (lines 316-323). In addition to multi-agent frameworks (PMC), we explicitly compared against popular sole-planning methods (Reflexion for monolithic planning and EvoAgent for multi-agent planning) by augmenting them with ReAct-based search (`Table 1`).
>
> **Regarding Hao et al. [1]**: The reason we didn’t include [1] in our baseline choices is not because it is "sandbox-based", but rather because of its assumption that is fundamentally impractical for real-world travel planning. Even if we provided [1] with the same searched data, their approach requires *prior knowledge of the evaluation metrics encoded in its prompts*, which contradicts our core settings for real-world travel planning:
> * Impractical constraint assumption: [1] does not dynamically discover constraints as in ATLAS. Instead, when translating user queries into code for formal solvers, it uses prompts with *hardcoded* constraints and hints on evaluation criteria. For instance, *“Assert first transportation happens at first day... get accommodation minimum night info and assert it to be less than the day's stay...”* (Appendix G.1.1 in [1]). This reliance on prior domain knowledge significantly restricts the method’s generalization to broader open-domain contexts where constraints are unknown.
> * Real-world constraint discovery: In a real-world setting, an agent does not start with a predefined list of types of constraints to look for. Relying on such benchmark-specific knowledge constitutes a form of evaluation metric leakage that simplifies the task from constraint discovery to merely variable filling.
>
> ATLAS is specifically designed to address the constraint construction challenge (Section 3.1): identifying what the constraints are from raw context without a pre-existing schema. Comparing ATLAS to a method that assumes the constraint types are known a priori would not be a valid evaluation of the agent's ability in a practical setup. We acknowledge that this is an important clarification, and have added this discussion to Appendix A.
>
> &nbsp;
> ## Lack of efficiency analysis (W3)
> We’d like to first clarify that Section 3.4 is intended to illustrate how our framework can naturally extend to support multi-turn travel planning efficiently, rather than to claim that "ATLAS is efficient due to information reuse." Regarding the cost analysis, we direct the reviewer to `Appendix D.4` (with a pointer in line 354) where we present a detailed analysis on runtime and token cost breakdown per module in our ATLAS framework, as well as a comparison of total cost across all baseline methods (`Table 11`).
>
> Additionally, we present the estimated monetary cost (output tokens) for the TravelPlanner validation set, based on standard pricing for Gemini-2.5-Pro ($10 per 1M output tokens). The table below compares the 25th, 50th (median), and 75th-quantile costs (in USD):
>
> | Method | 3-day plan | 5-day plan | 7-day plan |
> | :---: | :---: | :---: | :---: |
> | ReAct | (0.003, 0.011, 0.171) | (0.007, 0.014, 0.021) | (0.010, 0.018, 0.023) |
> | ReAct+Reflexion | (0.006, 0.008, 0.011) | (0.013, 0.016, 0.024) | (0.018, 0.020, 0.026) |
> | ReAct+EvoAgent | (0.065, 0.071, 0.079) | (0.159, 0.194, 0.218) | (0.202, 0.229, 0.261) |
> | PMC | (0.059, 0.067, 0.071) | (0.114, 0.127, 0.141) | (0.151, 0.166, 0.191) |
> | ATLAS | (0.015, 0.095, 0.205) | (0.055, 0.193, 0.630) | (0.053, 0.241, 0.439) |
>
> While ATLAS incurs higher costs than the single-agent ReAct baseline, this is the necessary trade-off for the massive performance gain (improving the final pass rate from 20.56% to 44.44% in Table 1). Furthermore, as noted in Appendix D.4 and the above table, ATLAS’s costs remain comparable to other multi-agent baselines for the majority of tasks while delivering superior results.
>
>
> &nbsp;
> ## Preference Constraints (W4)
> We highlight that we already evaluated ATLAS on the Flex-TravelPlanner benchmark [3].
> As detailed in `Section 4.2 (Table 2)`, we adopted their multi-turn setting where users introduce preferences that impact the plan either *locally* (e.g., ''I actually want to try Italian cuisine during the travel'') or *globally* (e.g., ''I want to adjust my budget down to $2000''). \
> Our results demonstrate that ATLAS significantly outperforms baselines in this dynamic environment.
> It achieves this by adaptively triggering targeted search to resolve any information gap caused by new preferences, and then applying minimal repairs (for local preference updates) or comprehensive revisions (for global preference updates).
>
> We also appreciate you bringing COMPASS [2] to our attention. As it is concurrent work released after the ICLR deadline, we could not include it, but our Live Search setup (Section 4.3) simulates a similar dynamic preference environment where users provide feedback iteratively (see Figure 4).

---

> ### Author Response · Authors · 2025-11-22
> **Response to Reviewer i4Up (3/3)**
>
> ## How often is interleaved search triggered (Q1)
> > INTERLEAVED SEARCH seems to play an important role, in what proportion of test cases was it actually triggered?
>
> We investigated and found that the Interleaved Search was triggered in 40% of the validation set (72/180) and 44.4% of the test set (444/1000).
>
> Here is a detailed analysis on the number of interleaved search steps triggered across the validation set with a maximum number of interleaved search steps set to 10.
>
> | # Interleaved Search Steps | # Tasks | Proportion | Interpretation |
> | :---: | :---: | :---: | :--- |
> | 0 | 108 | 60.00% | Initial search was sufficient |
> | 1–9 | 57 | 31.67% | Successful compensation. Initial search failed (unsat verdict) but the Search Advisor successfully guided interleave search to fix it. |
> | 10 (max) | 15 | 8.33% | Unrecoverable failure
>
> The above table reveals that for 31.67% of the tasks (1-9 steps), the initial search stage failed to provide sufficient information (triggering an unsat verdict).
> However, the Search Advisor in the later search stage effectively compensated for this earlier failure by performing targeted retrieval, allowing the Checker to eventually validate the plan (stopping before the limit).

---

### Official Review · Reviewer_W8Ma · 2025-11-01

**Soundness:** 3
**Presentation:** 3
**Contribution:** 3
**Rating:** 6
**Confidence:** 5

**Summary:**

This paper considers the travel planning problem which should follow the real-world constraints. Te authors present a multi-agent framework, ATLAS, to trackle three identified core challenges, constraint construction, constraints-aware answering, and resolving information gap. Specifically, they design a constraint management agent to identify and codify the constraints, a planner agent to propose a travel plan with the extracted constraints in a propose-validate loop, a search agent to diagnose failures and guide the further information collection. They conduct the experiments on the TravelPlanner benchmark and its multi-turn variants, to validate its effectiveness.

**Strengths:**

1. The most compelling aspect of this work lies in its decomposition of the travel planning problem into three well-justified challenges, a perspective with which I strongly agree, not only in the context of travel planning but also in broader real-world scenarios where agents must fulfill user needs.
2. Building upon the identified challenges, the authors have thoughtfully designed distinct agents that collaborate effectively to accomplish the complex and practical goal of travel planning.
3. The performance of ALTAS is significant and the effectiveness is well validated based on the ablation study.

**Weaknesses:**

I don't have major concerns on this work, here are some minor concerns.

1. It is worth noting that the experimental comparison in this paper does not include OpenAI's GPT models. Based on my own experiments with TravelPlanner, GPT-5 in sole-planning mode (i.e., planning under constraints as described in the paper) already achieves performance far exceeding the near-zero result reported for the original TravelPlanner, even approaching the performance of ALTAS in Table 2. Given that constraint extraction in TravelPlanner is relatively straightforward, the main challenge in a multi-agent system would still lie in planning under constraints. Therefore, I would be interested to see the results of ALTAS when integrated with GPT-5.

2. I agree that the insights and the framework proposed in this paper are generalizable to tasks like travel planning, which require both information gathering and strict constraint satisfaction to meet user needs. However, the validation in this work is limited to the travel planning scenario. Although the TravelPlanner benchmark has gained broad attention, its support for constraint satisfaction is confined to propositional logic, includes only a limited set of constraints, and the process of extracting constraints from its template-synthesized queries is relatively straightforward [1]. In real-world scenarios, users often express diverse needs that require first-order logic (FOL) for proper representation, thereby extending constraint extraction and validation into a combinatorial FOL space. From this perspective, if the authors intend to claim robust constraint satisfaction capability in the travel planning domain, it would be valuable to include evaluation on the ChinaTravel benchmark [2], which assesses first-order logic constraint satisfaction.

[1]. Large Language Models Can Solve Real-World Planning Rigorously with Formal Verification Tools.
[2]. ChinaTravel: An Open-Ended Benchmark for Language Agents in Chinese Travel Planning

**Questions:**

1. I'm not entirely clear about how the checker component operates in practice. Is it based purely on LLM self-verification, or does it adopt an "LLM-modulo" style approach that incorporates additional symbolic verification tools?
2. Could the proposed method be generalized to first-order logic, particularly how the constraint extraction and verification processes could be generalized?

---

> ### Author Response · Authors · 2025-11-22
> **Response to Reviewer W8Ma (1/2)**
>
> Dear Reviewer W8Ma,
>
> We thank the reviewer for the thoughtful assessment and for recognizing the effectiveness and the generalizability of our problem formulation and the significant performance of ATLAS. Here we provide answers to the reviewer’s questions.
>
> &nbsp;
> ## Results with GPT-5 (W1)
>
> We conducted additional experiments using OpenAI’s GPT-5 (reasoning effort set to "medium" for the following results as we didn’t find a significant difference between “medium” vs “high” in sole-planning). Due to the high cost and rate limits, we performed experiments on a randomized subset of the TravelPlanner validation set (30 samples: 10 each for Easy, Medium, Hard on 3-day itineraries). We used 3 check steps and 10 interleaved search steps (same as default ATLAS settings).
>
> The results are as follows (all in %):
> | Methods | Delivery | Commonsense (Micro / Macro) | Hard (Micro / Macro) | Final Pass |
> | :--- | :--- | :--- | :--- | :--- |
> | Sole-planning | 100.00 | 95.83 / 70.00 | 71.43 / 46.67 | 33.33 |
> | Planning with search (ReAct) | 100.00 | 89.58 / 43.33 | 50.00 / 46.67 | 26.67 |
> | Planning with search (ATLAS) | 100.00 | 93.33 / 56.67 | 67.29 / 60.00 | 49.00 |
>
> * Sole-Planning Baseline: We confirm your observation that GPT-5 achieves a decent baseline (33.33%) in sole-planning mode, far exceeding older models reported in the original travel planner and [1]. However, a 66.7% failure rate remains, indicating that strict constraint satisfaction is still a major challenge even for SOTA reasoning models as you noted.
> * The Search Bottleneck: Crucially, we’d like to highlight that when the model must search for information (ReAct) rather than having it provided (sole-planning), performance drops to 26.67%. This highlights the "Information Gap" challenge we posed: even a strong reasoner fails if it cannot retrieve the correct context.
> * ATLAS Improvement: ATLAS achieves a 49% final pass rate, significantly outperforming both baselines by leveraging two distinct structural advantages. First, it outperforms the sole-planning baseline, proving that our *multi-agent orchestration* handles complex constraints more effectively than a monolithic model, even when perfect search data is provided. Second, it substantially outperforms ReAct, demonstrating that interleaved search effectively tackles the information gap challenges. This confirms that ATLAS provides structural benefits complementary to the reasoning capabilities of SOTA models like GPT-5.
>
> [1] Planning with Multi-Constraints via Collaborative Language Agents. Zhang et al., 2025. https://aclanthology.org/2025.coling-main.672.pdf
>
> &nbsp;
> ## Generalization to First-Order Logic and Benchmarks (W2, Q2)
> > Could the proposed method be generalized to first-order logic, particularly how the constraint extraction and verification processes could be generalized?
>
> This is an excellent question that touches on the core design principles of our framework. We answer this by highlighting two principles behind our formulation:
> * (1) CSP as a general formulation: We define the problem as a CSP to provide a general abstraction. The choice of logic (propositional vs. FOL) is merely a choice of the *language* used to express the constraints ($C$).
> * (2) Natural Language (NL) as the logic layer: ATLAS utilizes NL to express constraints, which naturally encompasses the expressivity of FOL without mapping to a restricted symbolic schema. Since NL can express FOL primitives, such as quantifiers ("for all days", "there exists a restaurant") and complex predicates ("arrival city must match departure city"; a relation between variables), our extraction and verification processes remain the same regardless of logical complexity. In fact, the constraints in TravelPlanner already require reasoning that mirrors FOL, rather than simple propositional slot-filling. Their evaluation criteria require satisfying various constraints that are (1) relational: ``The trip must be a closed loop’’ (Day1_city == Day5_city); and (2) universal (Day$i$_attraction $\neq$ Day$j$_attraction, $\forall i \neq j$).
>
> Please refer to Appendix D.1 and Table 4 for detailed description of the TravelPlanner benchmark and their constraints.
>
> Regarding ChinaTravel Benchmark, we appreciate this suggestion. We agree that it is a valuable testbed for further validating ATLAS's flexibility in handling various types of constraints. We will follow up with the results on this benchmark shortly.

---

> ### Author Response · Authors · 2025-11-22
> **Response to Reviewer W8Ma (2/2)**
>
> ## Clarification on the checker component (Q1)
> > how the checker component operates in practice.
>
> To clarify, the Checker component in ATLAS operates via LLM-based verification, not a symbolic solver. We prompt the LLM with the current plan and the list of constraints identified by the Constraint Manager agent (in natural language). The LLM acts as a "soft" verifier, checking if the plan satisfies the description of the constraints (please refer to `Appendix F.1` for the prompt used, and `Appendix E.1.1` for example outputs).
>
> &nbsp;
> **In relation to LLM-Modulo approaches:**\
> We highlight a fundamental difference between ATLAS and existing LLM-modulo approaches like [1], which helps explain our design choice.
> * Hao et al. [1] relies on translating queries into code for solvers (e.g., Z3) using prompts that require **prior knowledge of evaluation metrics and constraints**. For instance, their prompt explicitly instructs: *“Assert first transportation happens at first day... get accommodation minimum night info and assert it to be less than the days stay...”* (Appendix G.1.1 in [1]). We note that it is **not** practical to assume that we would know what constraints and evaluation criteria to be satisfied beforehand.
> * On the contrary, ATLAS operates without prior knowledge of metrics or constraints (lines 727-728). As part of the Constraint Construction challenge (Section 3.1), our Constraint Manager dynamically identifies constraints from the user query and search results. This allows ATLAS to handle open-world requirements (including implicit commonsense rules) that extend beyond the rigid capabilities of a predefined solver schema.

---

> ### Comment · Reviewer_W8Ma · 2025-11-22
>
> Thanks for your clarification.
>
> **About the W1**
> Due to the existing information gap, GPT-5 does not show a significant advantage over Gemini-2.5 Pro as reported in the paper. I agree that it is no longer necessary to include the GPT-5 results in the manuscript.
>
> **About the Q1 [solved]**
>
> **About the W2 and Q2**
> I agree that, using language as the medium, the method can be extended from propositional logic to first-order logic, although the specific performance after such an extension remains unclear.
>
> The two examples `closed-loop trip` and `avoid repetition` are still propositional constraints.
> **The distinction between propositional logic and first-order logic does not lie in the specific semantics of a constraint predicate, but rather in how the system combines, reasons, and determines validity concerning these predicates.**
> - Propositional logic: reasons about truth relations between whole propositions without examining their internal structure.
> - First-order logic: introduces domains, predicates, relations, and quantifiers, enabling expressive reasoning over structured objects.
>
> To support relational constraints and universal quantification, a system must allow arbitrary relations and functions defined over an object domain. If we take “daily activities” as the domain, an FOL-capable system could express constraints such as:
> $$lunch-cuisine(day_1-plan)\neq dinner-cuisine(day_2-plan)$$
> > Meaning: the lunch cuisine of the restaurants visited on day 1 must differ from that of dinner in day 2.
>
> $$\forall d1​\forall d2​, d1​=d2​→ attractions(d1​-plan) \cap attractions(d2​-plan) = \emptyset $$
> > Meaning: for all days of the trip, avoid revisiting attractions.
>
> Both the temporal function or the cuisine function can vary arbitrarily in user requirements, like `transport-type`, `breakfast-cuisine`, which FOL naturally supports as a compositional space of constraints.
>
> Supporting only propositional logic rather than first-order logic makes the problem resemble a pure constraint-satisfaction task, much like a Sudoku puzzle with more complex information, rather than a desirable real-world AI assistant.
>
> Overall, this does not diminish my evaluation for the work’s contribution to decomposing the challenges of the travel-planning problem (as I stated in my review), but rather share some thoughts with the authors.

---

> > ### Author Response · Authors · 2025-11-29
> >
> > We sincerely appreciate the reviewer for this prompt engagement and thought-provoking discussion.
> > We fully agree with your distinction: for an agent to be a desirable real-world assistant, it must be able to reason over constraints defined as arbitrary functions and relations over object domains, operating within a compositional space of requirements.
> >
> > As the reviewer noted, our CSP formulation serves as the structural abstraction for the interactions between agents, rather than a restriction to a specific logical formalism. Because our current instantiation of ATLAS uses Natural Language as its representation layer, it conceptually handles these arbitrary relations without architectural issues; our Checker (an LLM) naturally parses and reasons about properties as functions of event objects, effectively operating in that compositional space.\
> > That said, we agree that your insight regarding FOL as a formal representation is a valuable consideration for the future extension of ATLAS. While our current NL-based Checker is flexible, a neuro-symbolic instantiation would offer rigorous guarantees. For instance, the Constraint Manager could extract dynamic constraints (constructing the object domain), and a dedicated translation module could convert these into formal FOL representations. This would allow us to augment the planner with a formal solver, combining the flexibility of LLMs with the reliability of symbolic logic.
> >
> > We note that constraints similar to the FOL examples you specified (i.e., specific cuisine preferences or avoiding repeated attractions) are indeed part of the evaluation metrics in TravelPlanner, and ATLAS’s strong performance there hints at its capability to handle such constraint types. However, we agree that extending the evaluation to benchmarks with more vast combinatorial space of real-world travel constraints is truly valuable.

---

### Meta-Review · Area_Chair_P5xj · 2026-01-04

**Summary:**

Contribution Summary:

This paper proposes ATLAS, a multi-agent collaboration framework formalized as a dynamic Constraint Satisfaction Problem (CSP). It introduces mechanisms for constraint management, iterative plan verification, and adaptive interleaved search to effectively handle complex implicit constraints in real-world travel planning, significantly outperforming existing baselines on the TravelPlanner benchmark.

Reviewer Concerns:
* **Novelty and Formalization:** Reviewers questioned whether the framework was largely a "repackaging" of existing multi-agent/verification ideas (e.g., CRITIC, PMC) and noted that the connection between the theoretical CSP formalization and the actual system implementation appeared superficial or arbitrary.
* **Baseline Comparisons:** Several reviewers noted the absence of comparisons against stronger or more recent baselines, specifically OpenAI's GPT-5 (to test if advanced single-agent reasoning suffices), Hao et al. (2025), and other LLM-Modulo or verifier-aided approaches.
* **Generalization and Scalability:** Concerns were raised regarding the evaluation being predominantly limited to the TravelPlanner benchmark (raising overfitting risks), the lack of validation on complex First-Order Logic (FOL) constraints, and the heavy reliance on manual prompt engineering which may hinder scalability.
* **Analysis and Efficiency:** Reviewers requested more granular analysis, including failure breakdowns per subtask (e.g., search vs. planning success rates), a quantitative cost/efficiency analysis, and clarification on counter-intuitive results where performance on "Hard" subsets exceeded "Medium" subsets.

**Reviewer Concerns:**

Addressed Concerns:
* **Lack of State-of-the-Art Baseline (GPT-5):** The authors conducted new experiments with GPT-5. The results showed that ATLAS significantly improves GPT-5's performance (from ~33% in sole-planning to 49% with ATLAS), proving the framework's value beyond just the base model's capability. Reviewer W8Ma explicitly accepted this result.
* **Novelty and System Design:** The authors provided a detailed justification mapping their agents to standard CSP components (e.g., Search Agent = Domain Construction; Constraint Manager = Constraint Acquisition) and provided ablation studies proving the necessity of specific agents. Reviewer Qh1s found this clarification sufficient to resolve their concerns about "arbitrary" design.
* **Granular Analysis:** In response to Reviewer i4Up, the authors provided a detailed component-level breakdown (isolating Search vs. Planning failures) and a cost analysis table. This quantified the value of the "Interleaved Search" mechanism.
* **Missing Baselines (Hao et al.):** The authors argued convincingly that Hao et al. relies on metric leakage (hardcoded constraints in prompts) which is inapplicable to the open-world setting of ATLAS.
* **Data Anomalies:** The authors successfully explained the "Hard > Medium" performance phenomenon as a result of stricter constraints reducing the search space, which satisfied Reviewer LbP3.

Outstanding Concerns:
* **Evaluation on FOL Benchmarks (ChinaTravel):** While the authors argued that their NL-based approach covers First-Order Logic implicitly and promised future results on the ChinaTravel benchmark, these results were not part of the rebuttal data. However, Reviewer W8Ma noted this was not a major blocking issue.
* **Manual Prompt Engineering:** Reviewer LbP3's concern regarding the scalability of manual prompt engineering remains valid, as the authors only suggested future automation (e.g., DSPy) rather than a current solution.

**Reviewer Scores:**

* **Reviewer W8Ma:** Original Score: 6. Likely Final Score: 6.
   The reviewer originally rated it "marginally above the acceptance threshold". Their main request was comparing the method against GPT-5. After the authors provided these results showing ATLAS outperforms GPT-5, the reviewer explicitly stated they "agree that it is no longer necessary to include the GPT-5 results" and reiterated that they "don't have major concerns".
* **Reviewer Qh1s:** Original Score: 4. Likely Final Score: 6.
   The reviewer originally rated it "marginally below". Following the rebuttal which clarified the novelty and CSP formalization, the reviewer explicitly commented: "I think the author has well resolved my confusion, and I will raise the score... I support accepting this paper".
* **Reviewer i4Up:** Original Score: 6. Likely Final Score: 6.
   The reviewer originally rated it "marginally above" but requested component-level failure analysis and cost efficiency data. The authors provided the exact ablation tables and cost breakdowns requested.
* **Reviewer LbP3:** Original Score: 8. Likely Final Score: 8.
   The reviewer was originally very positive, rating it an "accept". Their only question regarding the counter-intuitive performance on "Hard" vs "Medium" datasets was satisfactorily explained by the authors as a characteristic of the benchmark's constraint space.

---

### Decision · Program_Chairs · 2026-01-26

Accept (Poster)